# Significance and Distribution of Apatite in the Triassic Doig Phosphate Zone, Western Canada Sedimentary Basin

**Pablo Lacerda Silva \*** and **R. Marc Bustin**

Department of Earth, Ocean and Atmospheric Sciences, The University of British Columbia, Vancouver, BC V6T 1Z4, Canada; bustin@mail.ubc.ca
\* Correspondence: pablols@alumni.ubc.ca; Tel.: +1-604-442-4983

**Abstract:** The Doig Phosphate Zone (DPZ) is a phosphate-bearing marine unit located at the base of the Doig Formation, in the Triassic section of the Western Canada Sedimentary Basin. The DPZ has a maximum thickness of 90 m and extends across northeastern British Columbia and west-central Alberta. In this study, we characterize the significance and interpret the origin of apatite in the DPZ through mineralogical and geochemical analyses, thin section study, and field emission scanning electron microscopy. The occurrence of apatite in the DPZ is not evenly distributed but restricted to discrete 10 to 20 cm thick beds, located near the base of the DPZ. Phosphorites are of two types: grainstones composed primarily of unconformity-bounded coated grains, and intraclastic phosphorites composed of detrital silt-sized grains and apatite coated grains in a cryptocrystalline phosphatic matrix. The phosphorite beds are records of stratigraphic condensation due to low detrital input during transgression. The erosionally truncated phosphatic coated grains and intraclasts are interpreted to be a result of various phases of phosphatization, exhumation, erosion, reworking, winnowing, and redeposition in alternating quiescence and storms or bottom currents. The abundance of pyrite and chalcophile trace elements, as well as the low concentration of proxy elements for organic matter productivity and preservation, are further evidence of stratigraphic condensation, with sulfidic pore water development and extensive organic recycling promoted by biological activity during the long exposure times. The phosphorites were formed under oxygenated water conditions, as suggested by the depletion in Ce and the presence of a diverse benthic fauna.

**Keywords:** Doig Phosphate Zone; phosphorite; apatite; coated grains

## 1. Introduction

Sedimentary phosphorite deposits, although uncommon, occur around the world throughout the Phanerozoic rock record [1,2]. The formation of sedimentary phosphorites is generally attributed to a unique confluence of processes and conditions that allow phosphorous to be concentrated, deposited, and preserved [3]. These episodic phenomena are affected by global cyclical patterns of atmospheric circulation, eustatic sea-level, and landmass distribution, but also respond to short term changes in these patterns, as well as local variations in phosphorus flux, bioproductivity, water and sediment column stratification, and detrital sedimentation rate [2].

Phosphorus is present in minor amounts in most rocks, but phosphorites ($P_2O_5$ exceeding 18 wt.%) may reach values in excess of 40 wt.% [4,5]. Föllmi et al. [6] classified sedimentary phosphorites into pristine, condensed, and allochthonous. Pristine phosphorites show no evidence of reworking and are interpreted as the product of in situ formation by primary phosphogenesis, which involves the direct precipitation of apatite within sediments near their interface with water or during diagenesis.

Condensed phosphorites are phosphorite beds and laminae of phosphate grains concentrated by winnowing and bioturbation. Allochthonous phosphorites are composed of phosphatic grains transported and redeposited by turbulent or gravity flows.

The Doig Phosphate Zone (DPZ) is an informal, laterally extensive and variable phosphorous-bearing, unit defined by Creaney and Allan [7] at the base of the Doig Formation, in the Middle Triassic section of the Western Canada Sedimentary Basin (WCSB). The DPZ provides an opportunity to understand the role of phosphogenesis in the sedimentation on the western margin of the WCSB, as well as the general processes and depositional environments of sedimentary phosphate deposits. Furthermore, the phosphate deposits of the Toad Formation, the stratigraphic equivalent of the DPZ in outcrop, represent the most promising phosphate resources and associated elements such as V, As, U, Y, and REE (rare earth elements) in British Columbia [8]. In this study, we characterize the significance and interpret the origin of apatite in the DPZ. We describe and quantify the abundance, mode of occurrence, and stratigraphic and spatial distribution of phosphate and associated elements. The apatite-rich beds and associated strata within the DPZ are described here in terms of their bulk mineralogy and major elements of lithogeochemistry. The trace element geochemistry of these samples is investigated, with a focus on the preferential incorporation of elements from the seawater during phosphogenesis, reflected by paleodepositional and early diagenetic conditions [9]; and the role of paleoredox conditions on the accumulation and preservation of organic matter. The mode of occurrence of apatite is further examined in detail through petrographic thin sections and scanning electron microscopy imaging.

Previous studies have made important contributions to the study of the DPZ regarding its age, depositional environment, and provenance [10,11], as well as the refinement of its sequence stratigraphic framework and organic matter distribution [12,13]. Studies on the sedimentology, mineralogy, textural aspects, composition, and the stratigraphic and spatial distribution of apatite and phosphorite beds in the DPZ are, however, notably absent from the literature. Furthermore, subsurface mapping of the DPZ using well logs is challenging due to the difficulty in determining both the top and base across the basin. The precise definition of the DPZ top is uncertain because of the often interbedded nature of phosphate-rich beds with siltstone beds and the informally named Gordondale sandstone [14]. The basal contact is sharp and its definition is less problematic, but legacy mapping often included the Sunset Prairie Formation [15] in British Columbia (BC) or the Lower Doig Siltstone (LDS) in Alberta [14], in the DPZ.

*Geology*

The DPZ corresponds to the base of the second cycle in a series of three major third or fourth-order transgressive-regressive cycles that comprise the Triassic strata in the WCSB [16,17]. The phosphate zone has been considered a condensed section formed during transgression [16]. During the Triassic, the WCSB was located at the northwestern margin of the Pangea supercontinent, facing the open oceanic regime of the Panthalassa, at about 30° N of latitude [18]. As a result of the paleogeographical configuration, the sedimentary environments were dominantly fine-grained siliciclastics deposited on marine shelves and ramps with associated nonmarine aeolian and evaporitic settings with low fluvial input.

The DPZ is an organic-rich, radioactive dark mudstone and argillaceous siltstone with common phosphate granules and nodules, interbedded with calcareous siltstone and dark-grey shale [19]. The DPZ is distinguishable in well logs by its high gamma-ray signature, and it is lithologically distinct from the upper section of the Doig Formation, which consists of mudstone, siltstone, and coquina beds with subordinate sandstone [20]. The Doig Formation, and the DPZ by extension, occur entirely in the subsurface, in the undeformed portion of the WCSB. The DPZ is considered a good to excellent hydrocarbon source rock comprised of Type II oil and gas-prone kerogen with total organic carbon (TOC) values ranging from 1.8% to 11% in weight [21,22], and is mature with respect to hydrocarbon generation across the entire northwestern portion of the WCSB [21,23]. Overlying Triassic conventional reservoirs of the Halfway, Charlie Lake, and Doig Formations, were sourced at least in part from

the DPZ, based on the correlation of biomarkers between kerogen from the DPZ and migrated petroleum [7,23,24].

## 2. Materials and Methods

Cored intervals from the DPZ in five wellbores (Table 1), covering a significant portion of the lateral extent of the Doig Formation subcrop area in British Columbia (Figure 1), were logged for lithology and sedimentary structures. From these intervals, 53 samples were selected for mineralogical and geochemical analyses, and a subset of these samples was selected for petrographic and scanning electron microscopical studies.

**Table 1.** Summary of core intervals logged and the number of samples analyzed per wellbore.

| Unique Well Identifier | Core Interval Logged (m) | Number of Samples |
|---|---|---|
| 100/04-09-084-22W6/00 | 1631–1660.35 | 11 |
| 200/c-082-F 094-H-01/00 | 1045–1063 | 16 |
| 100/12-04-086-20W6/00 | 1593–1611.2 | 6 |
| 200/a-063-A 093-P-09/00 | 2420–2435.8 | 9 |
| 100/15-34-080-18W6/00 | 2045–2077.38 | 11 |

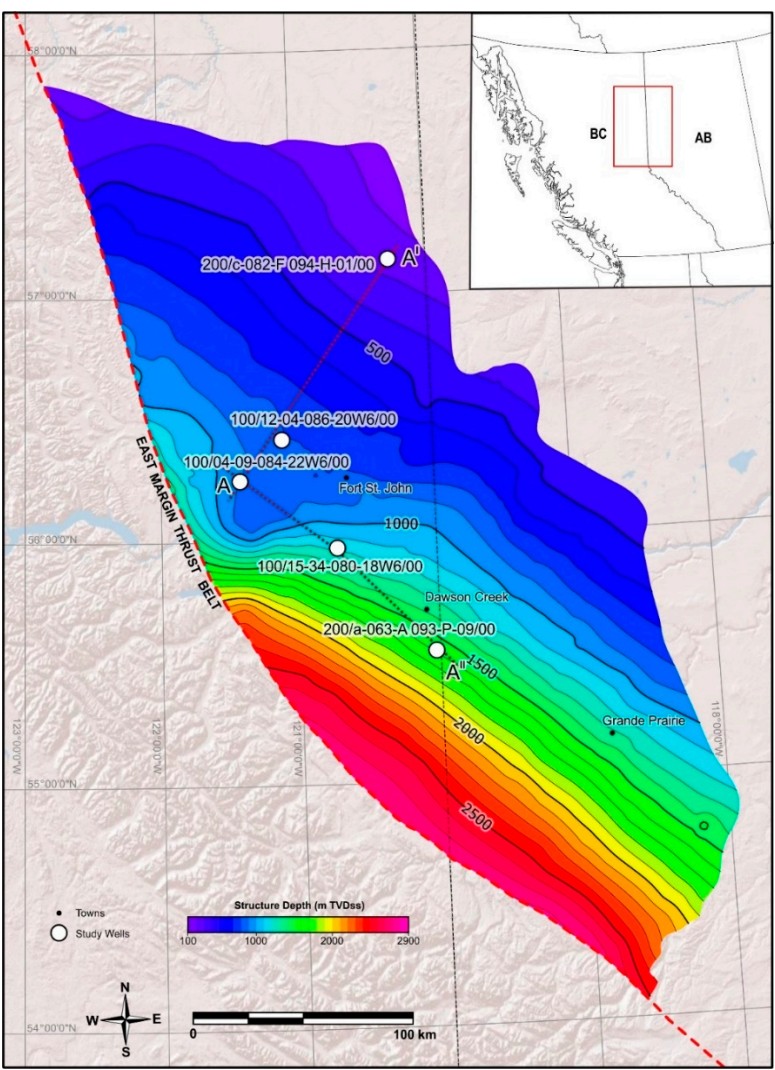

**Figure 1.** Location of wellbores used in this study and cross-sections on a map of the Doig top structure [25] with elevations expressed in vertical meters below mean sea level permanent datum, and shaded relief topographic backdrop map.

## 2.1. X-ray Diffraction

Apatite abundance was obtained from bulk mineralogy analyses by X-ray diffraction (XRD), according to the method outlined by Munson et al. [26]. The analysis was performed using normal-focus CoKα radiation on a Bruker® D8 Focus diffraction system at the Earth, Ocean and Atmospheric Sciences Department of the University of British Columbia in Vancouver, BC, Canada, with diffraction patterns obtained over the range of 3–70° 2θ at a step size of 0.03° and 0.8 s. Proportions of the mineral phases were quantified using the Rietveld [27] method of diffraction full-pattern fitting on the Bruker® AXS Topas V3.0 software.

## 2.2. Geochemistry

Geochemical analysis was carried out on 2.5 g of pulverized material under 75 microns, by a combination of inductively coupled plasma-atomic emission spectrometry (ICP-AES) inductively coupled plasma–mass spectrometry (ICP-MS), LECO furnace combustion with infrared spectroscopy, and loss on ignition (LOI) on whole-rock samples, for a comprehensive characterization of major (greater than 1% by mass), minor (between 0.1 and 1%), and trace (less than 0.1%) elements. The geochemical analyses were conducted at ALS Geochemistry in North Vancouver, BC, Canada. Oxides of major rock-forming elements are analyzed by ICP-AES and reported as a percentage of oxides, which are calculated from the determined elemental concentration. Base metals, trace elements, carbon, and sulfur are reported in parts per million (ppm) of their elemental state, and are measured at different determination ranges according to the method used. The geochemical analyses are for bulk element concentrations, although some elements may exist in multiple oxidation stages, such as sulfur occurring as sulfide or sulfate.

Limitations of ICP-AES prevent the measurement of some elements at trace concentrations, such as heavy alkali metals Rb and Cs, which are too readily ionized, as well as U, Th, W, and Ta, which are below the detection limits [28]. ICP-MS addresses some of the shortcomings of ICP-AES by offering extremely low detection limits and fewer interferences for REE and high-field strength incompatible elements. Comprehensive reviews of ICP-AES and ICP-MS methods are provided in Rowland [29] and Walsh [28]. Organic carbon was determined by Rock-Eval type pyrolysis on companion samples of each of the ICP-AES and ICP-MS samples, consisting of 70 mg of powdered material. Pyrolysis was conducted according to the method outlined in Espitalié et al. [30], and standard Rock-Eval parameters, including total organic carbon (TOC) and hydrogen index (HI), were calculated according to methods discussed by Peters [31].

### Control Samples

Sample contamination is a major concern in trace element analysis [32], and as the bulk samples had been previously crushed with a steel mortar and pestle, two samples were used to assess the contamination from the steel. One of the control samples consisted of a clear inclusion-free quartz crystal, while the other was a twin of one of the 53 samples. Each one of the control samples was split in two. One half was prepared in the same steel mortar and pestle, while the other half was prepared in an agate mortar and pestle. The comparison between the two sets of control samples prepared with the agate and the steel mortar and pestle suggests there is no significant contamination of major or trace elements, with the possible exception of Cu. The quartz crystal sample prepared with the agate equipment is composed of 99.4% $SiO_2$, while the one prepared with steel is 99.2%.

Among the major elements, the only significant difference between the agate and steel for the quartz crystal is in Fe, which is 0.2% for the agate versus 0.8% for steel. However, the phosphorite sample has no significant difference in Fe between the two materials. As for trace elements, Cu is significantly higher in the sample prepared in steel equipment for both the phosphorite, where it increases from 12 in agate to 32 ppm in steel, and the quartz crystal, where it increases from 2 to 15 ppm.

Other elements common in steel, such as C and Mo do not show significant discrepancies between the agate and steel mortar and pestle for neither the phosphorite nor the quartz crystal samples.

The difference in Cu between the samples prepared in agate versus steel equipment is attributed to heterogeneity between samples, as discrepancies in other trace elements unrelated to steel composition suggest. As for the potential contamination suggested by the higher amount of Fe from the mortar and pestle in the quartz crystal sample prepared with steel tools, the same effect is not observed in the phosphorite samples. This implies that the contamination observed in the quartz sample is an overestimation due to the highly abrasive nature of the quartz crystal during the sample grinding, which does not occur in the softer rock samples.

### 2.3. Petrography

Petrographic thin sections were used to investigate the textural aspects of apatite occurrence under transmitted light. The thin sections prepared with a blue dye epoxy to highlight porosity, and were double-stained for the identification of carbonate minerals and feldspars. The carbonate staining is a mixture of the organic dye Alizarin Red-S (ARS) and potassium ferricyanide ($K_3[Fe(CN)_6]$) dissolved in a dilute hydrochloric acid solution [33–36]. The feldspar staining consists of hydrofluoric acid vapor etching, immersion in barium chloride solution, rinsing, and treatment with a solution of potassium rhodizonate combined with cobaltnitrite [37].

### 2.4. Scanning Electron Microscopy

Field emission-scanning electron microscopy (FE-SEM) imaging was conducted on an FEI Helios NanoLab™ 650 at the Centre for High-Throughput Phenogenomics of the University of British Columbia, in Vancouver, BC, Canada. Two samples, representing different modes of apatite occurrence, were selected for FE-SEM. A billet of 2 by 2 cm was cut from each sample, the surface of which was polished with a polishing solution down to 0.05 micrometers. The samples were coated with carbon and images were acquired with the Everhart-Thornley (ET) detector in backscattered electrons (BSE) mode with a 20 kV accelerating voltage and 3.2 nA beam current. Elemental mapping and spot analysis were carried out on all images using the attached high-throughput silicon drift detector energy dispersive X-ray spectrometer (SDD-EDS) EDAX TEAM™ Pegasus system.

### 3. Results

The DPZ is mappable throughout the entire extension of the Doig Formation, based on its distinctive gamma-ray and uranium from spectral gamma-ray log character. The interval ranges from approximately 90 m thick in two depocenters adjacent to the east margin of the fold and thrust belt; one located in the deepest part of the basin south of Dawson Creek, and another in the Hudson Hope Low on the westernmost end of the Fort St. John Graben (Figure 2). The elevation below sea level of the DPZ structural top ranges from 150 m, near the northeastern subcrop edge, to 2850 m, near the southwestern deformation edge. Phosphorites, defined based on a $P_2O_5$ concentration higher than 18 wt.% [4,5], occur only at the base of the DPZ. Phosphorite beds were only identified in two wells (Figures 3 and 4), where the gamma-ray log reaches values upwards of 400 gAPI. In the remaining three wells analyzed, apatite is more concentrated near the base, but does not form phosphorite beds. Apatite occurs dispersed in variable concentrations throughout the upper intervals of the DPZ, causing a gamma-ray random sawtooth pattern with peaks exceeding 120 g API.

### 3.1. Sedimentology and Mineralogy of the Doig Phosphate Zone

The DPZ is comprised primarily of medium to light gray calcareous siltstone interbedded with 5 to 10 cm thick dark gray argillaceous mudstone beds (Figures 3 and 4). The siltstone and mudstone beds are generally parallel to wavy laminated, but subordinately cross-bedded in sandy siltstone beds, and were deposited in offshore to proximal shelf environments by suspension and occasional currents and gravity flow [20]. Horizontal and oblique burrows are common, and beds are often structureless

due to intense bioturbation. Calcareous siltstone also occurs as heterolithic linsen laminations within mudstone beds, and convolute lamination is rare. Bivalve fragments and occasionally gastropod fragments are present throughout the entire interval, mostly associated with argillaceous mudstones, but also occur in coquina beds of up to 10 cm thick. There are frequent calcite veins, nodules, and calcite-filled vugs, often near fossiliferous beds. The siltstones and mudstones are predominantly composed of quartz, calcite, and dolomite, with relatively low clay contents of up to 15% in weight. Calcite is present in the form of skeletal grains and cement. Plagioclase and K-feldspar content are variable and lower than 20% combined. Pyrite is an important accessory mineral, with a mean value of 3%, but reaching up to 15% in weight.

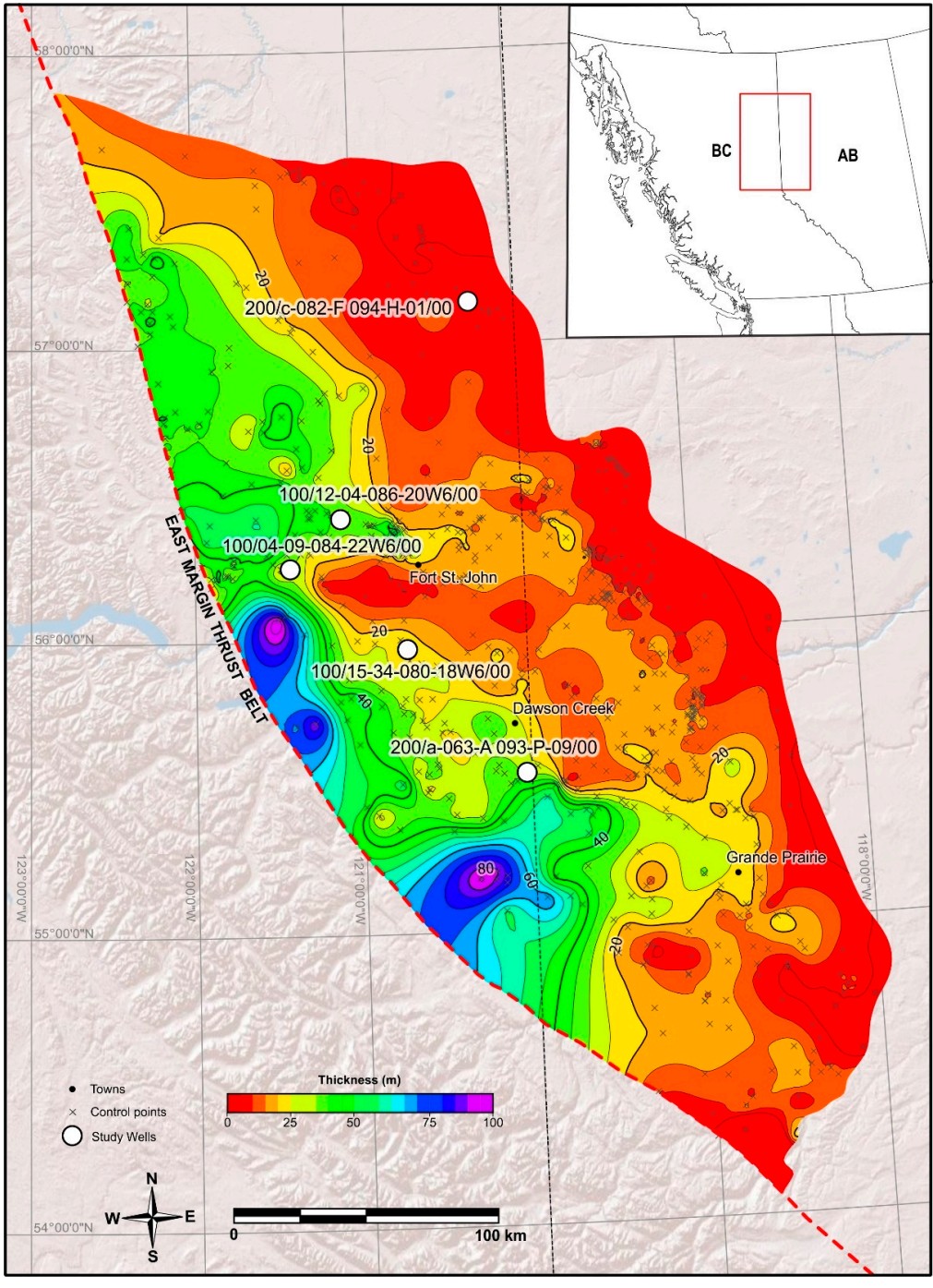

**Figure 2.** Isopach map of the Doig Phosphate Zone with thickness expressed in meters, location of well control points, and shaded relief topographic backdrop map.

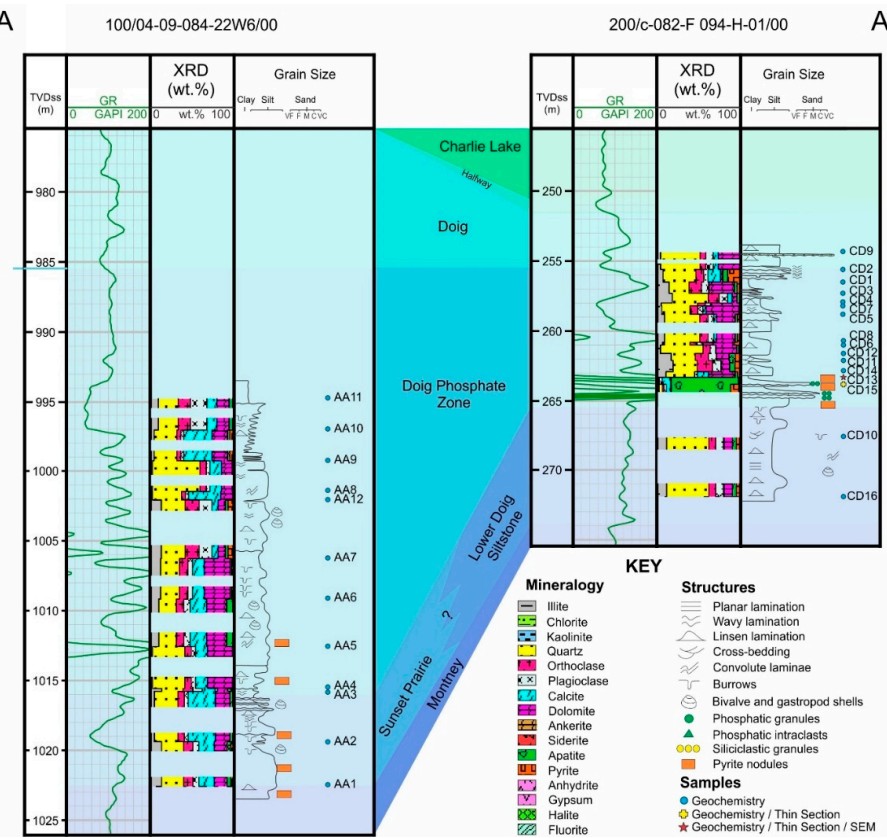

**Figure 3.** Stratigraphic dip cross-section of wells analyzed and described in this study, with gamma-ray log, core log, and sample locations. The location of the cross-section is shown in Figure 1.

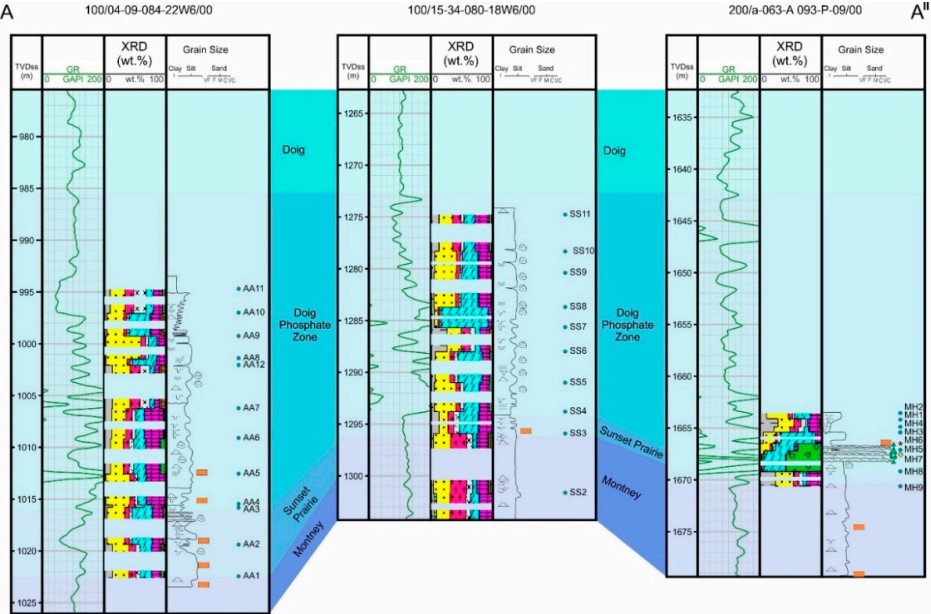

**Figure 4.** Stratigraphic strike cross-section of wells analyzed and described in this study, with gamma-ray log, core log, and sample locations. The location of the cross-section is shown in Figure 1 and the key is shown in Figure 3.

The DPZ contains apatite throughout, including in the siltstone and mudstone facies, where it occurs as sparse grains not readily identifiable in hand specimens, reaching up to 7% by weight. Phosphorite beds occur within the DPZ as discrete 10 to 20 cm thick moderately to poorly sorted

medium sand to granule and pebble-sized phosphate grain and intraclastic grainstone beds, with apatite content ranging between 40% and 80% by weight (Figure 5). Two types of phosphorites are distinguished based on the main type of phosphatic clasts; one facies is a moderately sorted phosphatic grainstone, composed primarily of phosphatic coated grains, while the second type is a very poorly sorted intraclastic phosphorite composed of granule to pebble-sized phosphatic intraclasts. Phosphorite grainstone beds are interbedded with medium gray, moderately to heavily bioturbated calcareous siltstone, with sharp and irregular contacts. Pyrite is common near the base of the DPZ, occurring as coatings on grains, siltstone lenses within mudstone beds, and nodules.

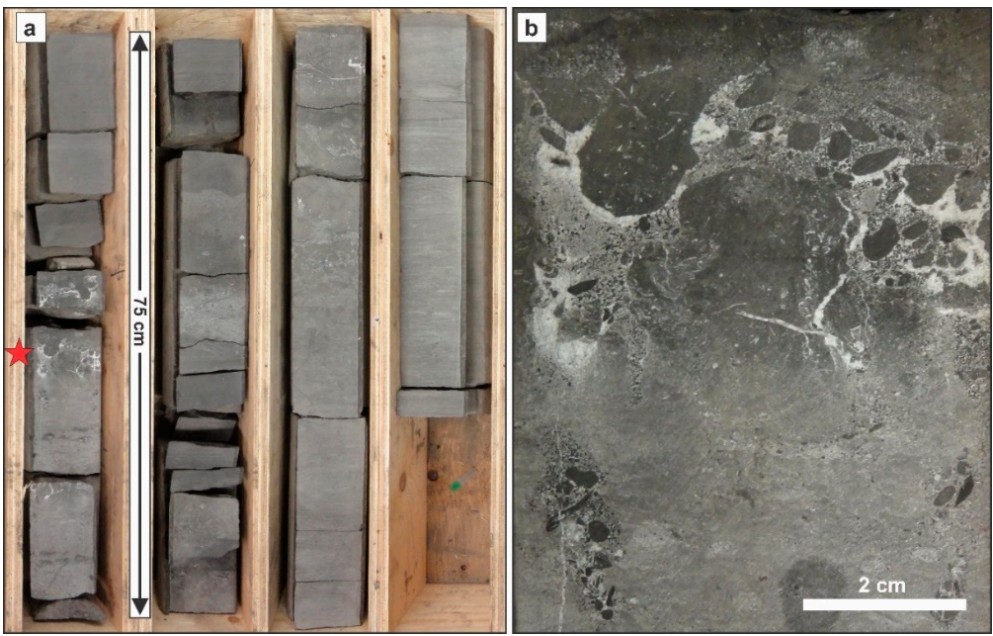

**Figure 5.** (**a**) Photographs of core box of the interval between 2422.46 and 2425.46 m of well 200/a-063-A 093-P-09/00, which contains a 15 cm intraclastic phosphorite bed, indicated by a red star, and corresponding to sample MH6. (**b**) Close up view of the phosphorite bed, showing the dark phosphatic intraclasts containing light skeletal carbonate fragments only partially phosphatized, and siliciclastic grains, embedded in a silt-sized phosphatic coated grains framework and cemented by calcite.

The phosphatic grainstone facies, observed in the well 200/c-082-F 094-H-01/00 (Figure 6a,b), is a moderately sorted granular phosphorite composed primarily of phosphatic coated grains measuring between 150 and 700 μm in diameter, but also containing up to 30% of completely phosphatized skeletal fragments and up to 10% of phosphatic intraclasts and pellets. The grains are cemented by calcite (Figure 6a), but locally the cement may be almost completely leached (Figure 6b). The phosphatic coated grains are elliptical to nearly spherical, and are mostly composed of multiple alternating cortices of lighter and darker-colored nearly opaque apatite in thin section. The skeletal fragments are composed of a diverse assemblage, including algal phytoclasts, ammonoid fragments, arthropod fragments, gastropods, foraminifera, and ostracods. Intraclasts are composed of quartz and feldspar silt-size detrital grains embedded in a disseminated cryptocrystalline phosphate matrix, and fragments of reworked granular coated grain phosphorites.

The facies observed in well 200/a-063-A 093-P-09/00 (Figure 6c,d) is a poorly to very poorly sorted granular phosphorite composed of granule to pebble-sized phosphatic intraclasts, in addition to phosphatic coated grains measuring between 100 and 300 μm in diameter, but reach up to 600 μm, pellets, and sparse phosphatized skeletal fragments. The phosphatic coated grains are elliptical to nearly spherical and most are composed of dark nearly opaque structureless apatite; however, some coated grains, commonly within phosphatic intraclasts, have visible cortices, which have been

partly replaced by calcite (Figure 6c). Sparry calcite cementation is pervasive and replaced apatite completely, which is evidenced by some pseudomorphs of phosphatic coated grains (Figure 6d). The skeletal fragments show signs of intensive reworking and the taxonomic groups are mostly unidentifiable. Intraclasts are composed of quartz and feldspar silt-size grains, apatite coated grains, and skeletal fragments partly replaced by calcite embedded in a disseminated cryptocrystalline phosphatic matrix. Signs of dissolution prior to the cementation and calcite replacement are evidenced by molds of completely dissolved coated grains and floating apatite grains.

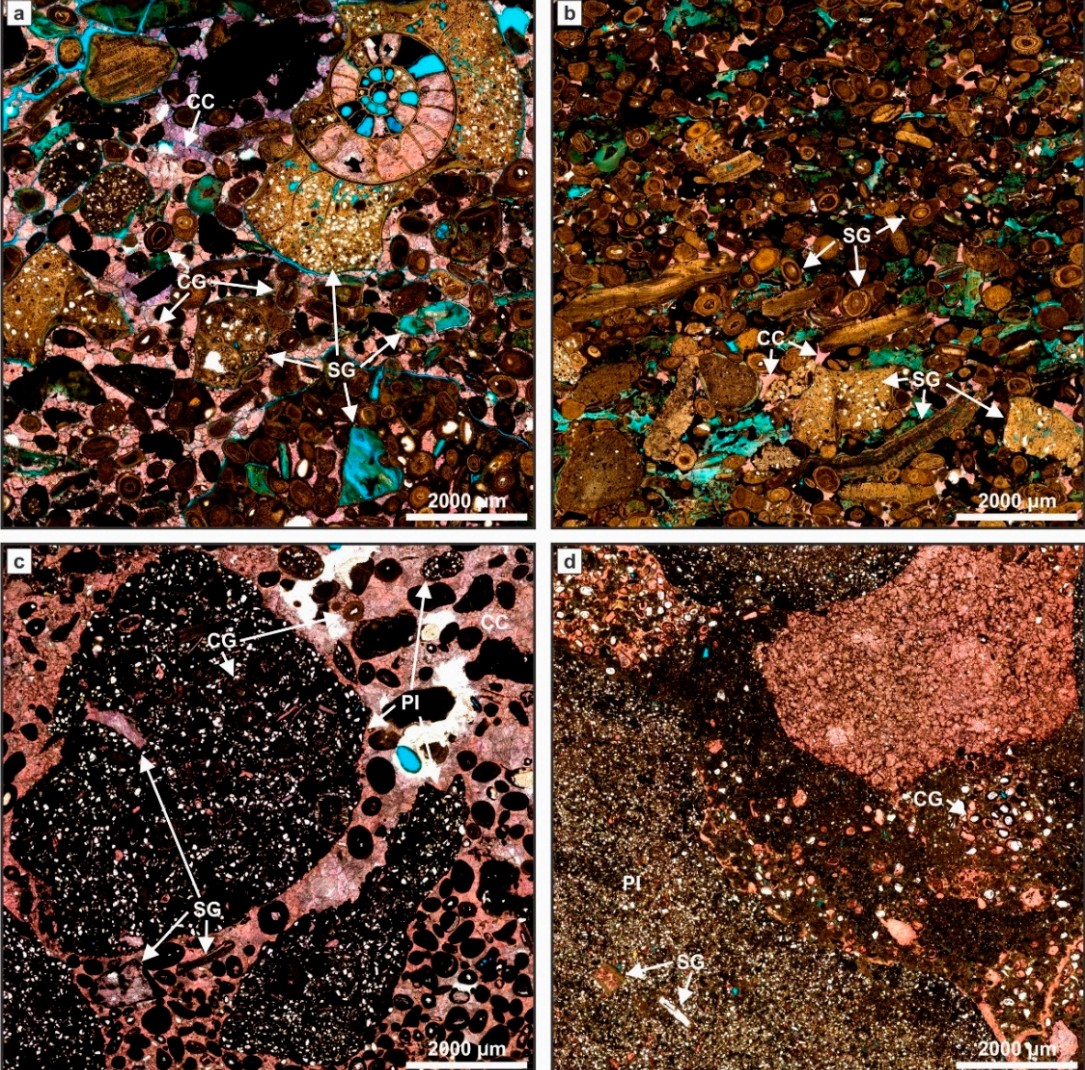

**Figure 6.** Photomicrographs of the phosphate-rich facies of the DPZ in plane-polarized light. (**a**) Sample CD13: phosphatized skeletal grains (SG), including ammonite in the upper-right corner, and coated grains (CG) cemented by red-stained calcite and purple-stained dolomite or ferroan calcite (CC), and predominantly intragranular or moldic porosity. (**b**) Sample CD15: Framework composed mostly of skeletal grains at the base of the photomicrograph, grading to predominantly coated grains at the top, with sparse calcite cementation and abundant intergranular, with subordinate intragranular and moldic porosity. (**c**) Sample MH6: two phosphate intraclasts (PI) and the edge of another one in the lower-left corner, containing silt-sized quartz and feldspar grains, partially phosphatized skeletal grains, and phosphate coated grains, with smaller intraclasts, skeletal and coated grains cemented by calcite (**d**) Sample MH7: the edge of a large phosphatic intraclast on the lower-left corner, containing abundant silt-sized quartz and feldspar grains, and rare partially phosphatized skeletal grains; in the upper-left corner and the right-hand side are two other intraclasts containing coarser detrital and coated grains, extensively cemented and replaced by calcite.

### 3.2. Structure, Composition, and Origin of Apatite Grains

The SEM imaging of apatite grains from two wells reveals compositional and morphological aspects of the various types of apatite grains found in the phosphorites. Most coated grains consist of irregular and often erosionally-truncated laminae, with cortices composed exclusively of apatite (Figures 7 and 8). This type of discontinuous grain was interpreted elsewhere by Pufahl and Grimm [38] as having formed through multiple phases of phosphatization, exhumation, erosion, and reburial into the zone of phosphogenesis (ZOP). The aforementioned authors described this grain type as unconformity-bounded (UB) and proposed that they represent different episodes of substrate reworking, winnowing, and redeposition caused by the alternation of quiescent periods with episodic storms or bottom currents.

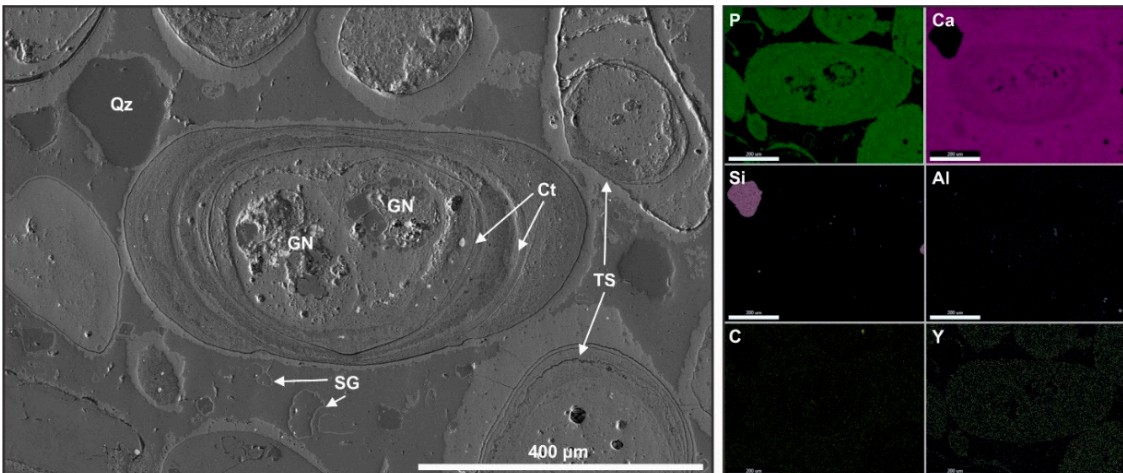

**Figure 7.** Detailed SEM image of sample MH6 centered on composite apatite coated grain, acquired with ET detector BSE mode, and SDD-EDS elemental maps of select major and trace elements. The grain in the center is formed by the agglutination of two apatite grains with aluminosilicate nuclei (GN) and multiple cortices (Ct) representing episodes of phosphatization. The grain on the bottom right corner has an internal truncation surface (TS) near the edge, and the grain on the upper right corner has an irregular shape with multiple internal, and an external truncation. Other grains are phosphatized skeletal fragments (SG) with moldic porosity later infilled by calcite, quartz (Qz), and aluminosilicates. All apatite grains and some silicates have a surrounding overgrowth rim of apatite. Cement is predominantly calcite. Yttrium occurs as a trace element homogeneously disseminated in phosphatic phases.

Due to the absence of significant quantities of vertebrate fragments or fish scales in the DPZ phosphorite beds, apatite was likely precipitated through sulfide-oxidizing microbial processing of phosphorous [39]. The high flux of organic matter due to upwelling and the hampered dilution by low detrital input caused by condensation would have enhanced the sulfide production by sulfate-reduction bacteria in the ZOP, creating favorable conditions for sulfide-oxidizing polyphosphate bacteria and the buildup of phosphate and fluoride in the pore waters (i.e., Föllmi et al. [6]). Apatite is thus interpreted as precipitating from pore waters supersaturated with phosphate a few centimeters below the sediment-water interface, where it nucleates around single detrital, skeletal, or organic particles, or grow around and embed multiple detrital grains.

Throughout the multiple erosion and deposition cycles, the position of the ZOP in the sediment column fluctuated up and down, causing growth in the form of cortices around coated grains; and if the ZOP is exposed to the sediment-water interface through removal of the overlying sediments by erosive currents, the grains would be truncated by reworking and concentrated by winnowing of finer particles. Phosphatization of carbonate skeletal grains is often partial and incomplete, which may be related to the migration of the ZOP by the removal of overlying sediments. Cementation by calcite

occurred early in the diagenetic sequence, as evidenced by the lack of mechanical compaction and floating grain contacts. The preservation of intergranular porosity in some granular phosphorite samples indicates a later phase of calcite dissolution.

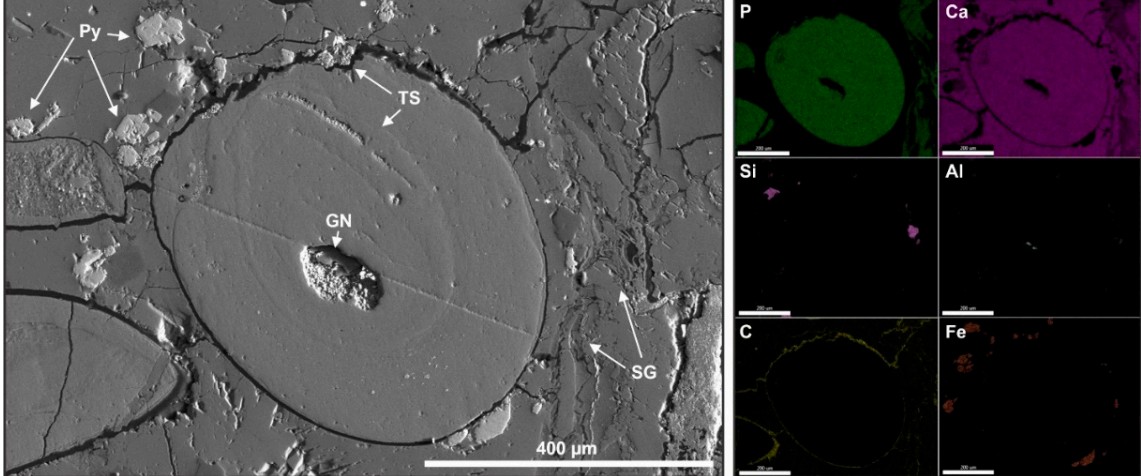

**Figure 8.** Detailed SEM image of sample CD13 centered on apatite coated grain, acquired with ET detector BSE mode, and SDD-EDS elemental maps of select major and trace elements. The grain in the center has an aluminosilicate nucleus (GN), and is compositionally homogeneous, but contains a subtle internal truncation surface (TS), as well as an external one. Apatite fragments on the right are partially phosphatized skeletal fragments (SG). Relatively large euhedral pyrite crystals (Py) are present near the edges of the coated grain. Cement is predominantly calcite, and solid bitumen is concentrated around grain contacts.

Phosphatized skeletal fragments with moldic porosity later infilled by calcite, as well as detrital quartz and aluminosilicate grains, occur in close association with coated grains. Apatite grains and sometimes silicate grains, often have apatite authigenic overgrowths. Cement is predominantly composed of calcite. The nuclei of coated grains are most commonly composed of aluminosilicates or quartz, and less commonly detrital organic matter (kerogen) or partially phosphatized calcareous skeletal fragments. The phosphatic grain cortices may have a uniform Ca-phosphate composition (Figure 8) or show a compositional banding of variable phosphate content (Figure 7). Truncation surfaces caused by reworking occur both internally to coated grains, and externally, in contact with the calcareous cement. Even in compositionally uniform grains, internal truncations caused by reworking are evident. Multiple episodes of reworking and phosphogenesis are also evident in grains with composite cores formed by the amalgamation of two apatite grains.

Yttrium, which serves as a general proxy for REE enrichment [40], is disseminated homogeneously in phosphatic phases, including coated grains and overgrowth rims (Figure 7). Solid bitumen, as detected by low density, habit, and the C signal from EDS, is concentrated as coatings on apatite grains and in fractures (Figure 8). Pyrite is present as relatively large euhedral crystals near the edges of coated grains (Figure 8) or small framboids of approximately 1 μm (Figure 9), frequently occurring in association with coated grains. The larger euhedral pyrite crystals are likely formed during later diagenesis [41], while the internal small framboids are associated with pyritization syngenetic with phosphatization [42].

The phosphatic intraclasts are composed of silt-size aluminosilicates and quartz grains embedded in apatite (Figures 10 and 11). Trace amounts of fluorine are detected and mapped in association with the apatite, which suggests that at least some are fluorapatite. Intraclasts often have an irregular but sharp contact, which can appear fuzzy by apatite overgrowths (Figure 10). Apatite rims also occur around other coated grains and siliciclastic grains. Multiple episodes of reworking are recorded by internal apatite mineralization rims, which is typically free from siliciclastic grains (Figure 11); the intraclast is later accreted by another layer of apatite with agglutinated siliciclastic grains. Solid bitumen is

concentrated around intraclast edges, but also occurs within intraclasts, around siliciclastic grains, likely adsorbed onto the crystal surface, as well as disseminated within the apatite. Pyrite is generally present as small crystals of approximately 1 μm, disseminated within intraclasts, or as larger euhedral crystals near internal intraclast borders.

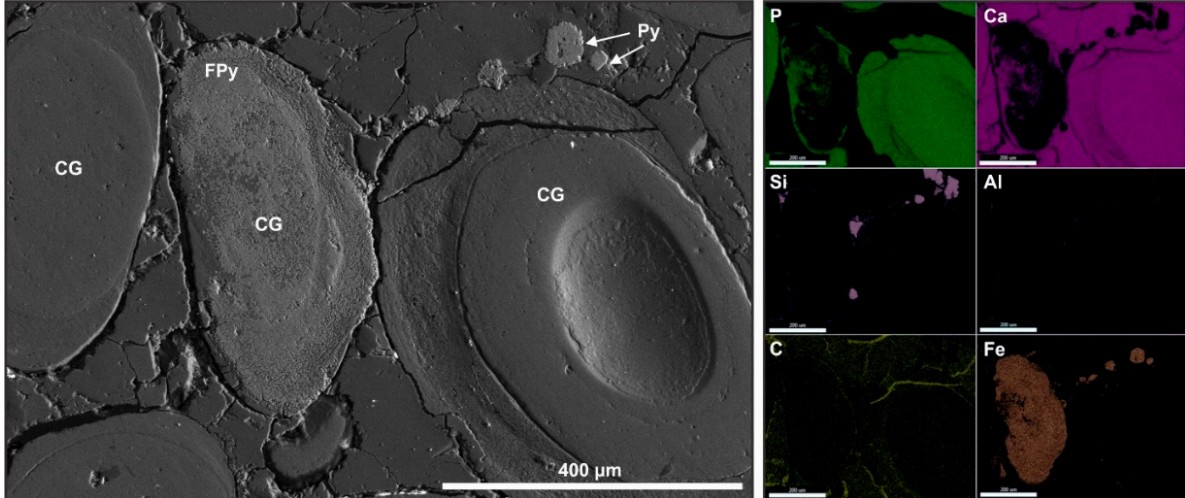

**Figure 9.** Detailed SEM image of sample CD13 centered on apatite coated grain (CG) with syngenetic framboidal microcrystalline pyrite (FPy), acquired with ET detector BSE mode, and SDD-EDS elemental maps of select major and trace elements. The grain to the left of the center has abundant internal framboidal pyrite, which also occurs around the upper edge of the grain to the right, interpreted as syngenetic. The larger pyrite crystal (Py) near the upper right corner of the image is likely formed during later diagenesis. Solid bitumen can be seen around grain contacts and in fractures.

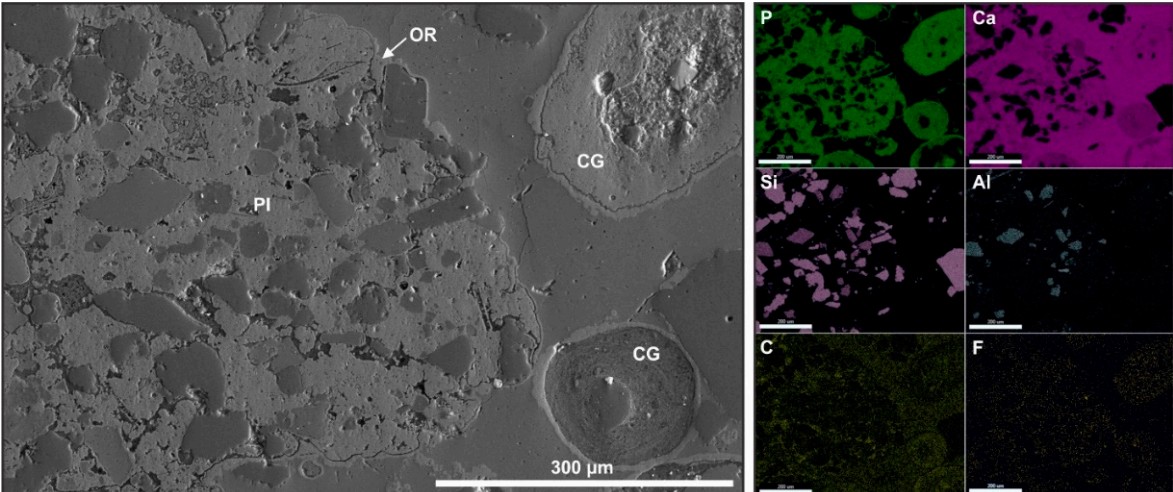

**Figure 10.** Detailed SEM image of sample MH6 centered between the edge of a phosphatic intraclast (PI) and two apatite coated grains (CG), acquired with ET detector BSE mode, and SDD-EDS elemental maps of select major and trace elements. On the left side of the image is the edge of a phosphatic intraclast containing aluminosilicate and quartz silt-size grains. Organic matter can be seen within the intraclast around siliciclastic grains, likely adsorbed onto the crystal surface. The intraclast has an irregular but sharp contact with an apatite overgrowth rim (OR). A rim can also be seen in the other coated grains and quartz grains in this picture. A trace amount of fluorine is mapped in association with the apatite, suggesting fluorapatite.

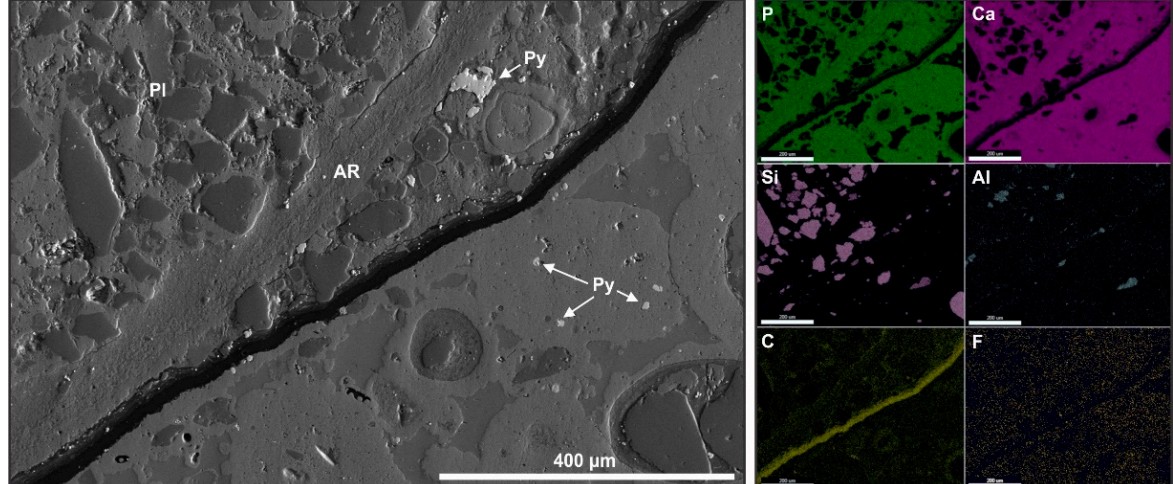

**Figure 11.** Detailed SEM image of sample MH6 centered on the contact of a phosphatic intraclast (PI) and the rock matrix and granular framework, acquired with ET detector BSE mode, and SDD-EDS elemental maps of select major and trace elements. The apatite intraclast includes silt-size aluminosilicates and quartz grains and records multiple episodes of reworking, as shown by the apatite mineralization internal rim (AR) without siliciclastic grains and the later episode of phosphatization and reincorporation of grains. On the outer edge, there is a large euhedral pyrite crystal (Py), as well as multiple smaller diagenetic pyrite crystals disseminated in the matrix. Solid bitumen is concentrated around the edge of the intraclast, but also occurs disseminated in apatite. A trace amount of fluorine is mapped in association with the apatite, suggesting fluorapatite.

## 3.3. Lithogeochemistry

### 3.3.1. Major Elements

Relative to the average concentrations of major elements in shales [43], the rocks from the DPZ are overall depleted in elements associated with siliciclastic facies (Si, Al, K, Na and Ti) and enriched in Ca and Mg. The Ca and Mg are associated with both carbonate bioclasts and cement. Relative to the concentration of 0.16 wt.% of phosphorus oxide ($P_2O_5$) in the average shale, all samples analyzed are enriched in $P_2O_5$, with a median value of 1.02 wt.%, and 90% of the samples having a concentration over 0.29 wt.% (Figure 12). The distribution of $P_2O_5$, however, is bimodal and its variance is one of the largest among the major elements, with the granular and intraclastic phosphorites containing up to 22.2 wt.% $P_2O_5$, while siliciclastic facies contain a background value of no more than 2.5 wt.%. The sulfur (S) content is significantly higher than that of the average shale, and iron (expressed as the oxide $Fe_2O_3$) has a wide range of values with a bimodal distribution. Organic carbon also has a wide distribution, ranging from less than 1 wt.% to nearly 8 wt.%. The kerogen is of Type II and Type III, with HI ranging from 18 to 325 mg HC/g TOC for thermal maturities ranging from $T_{max}$ values of 440 to 480 °C.

The correlation between all major elements was evaluated using Spearman's rank-order correlation coefficient ($r_s$), which enables the simultaneous comparison of strength and direction of all linear and non-linear monotonic relationships. Correlations are described as very strong for absolute $r_s$ values larger than 0.85, strong for $r_s$ values between 0.75 and 0.85, moderate for $r_s$ values between 0.5 and 0.75, and weak for $r_s$ values smaller than 0.5. Among the major elements, aluminum oxide ($Al_2O_3$) has the strongest positive correlations with potassium oxide ($K_2O$), with $r_s$ of 0.99, and titanium oxide ($TiO_2$), with $r_s$ of 0.97 (Figure 13). These elements are strongly associated with each other in the siliciclastic components of the sediments, particularly feldspar and illite. The strongest negative correlation of major elements is between CaO and $SiO_2$, followed by the other main elements associated with siliciclastic sediments, $Al_2O_3$, $TiO_2$, and $K_2O$. This is due to the low relative abundance of detrital grains in the phosphorites, which are Ca-rich due to the apatite and calcite cementation.

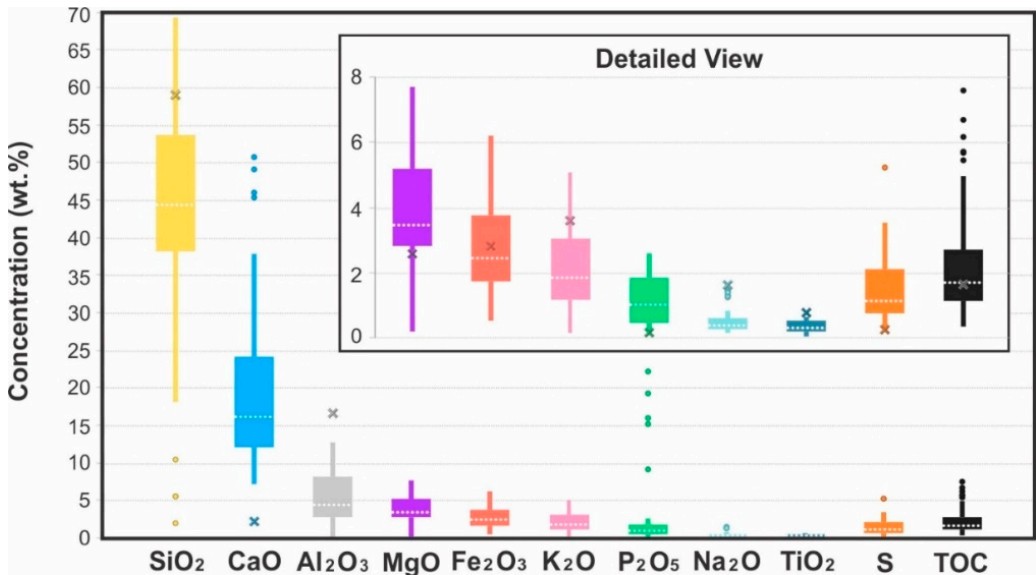

**Figure 12.** Box plot of all the major elements and TOC from Rock-Eval pyrolysis for all samples analyzed. The box represents the range between the upper and lower quartiles, the whiskers represent the lower and upper limit of adjacent values, and outliers are represented by dots. The dotted white line represents the median. Crosses represent the global average concentration of elements [43] and TOC [44] in shales.

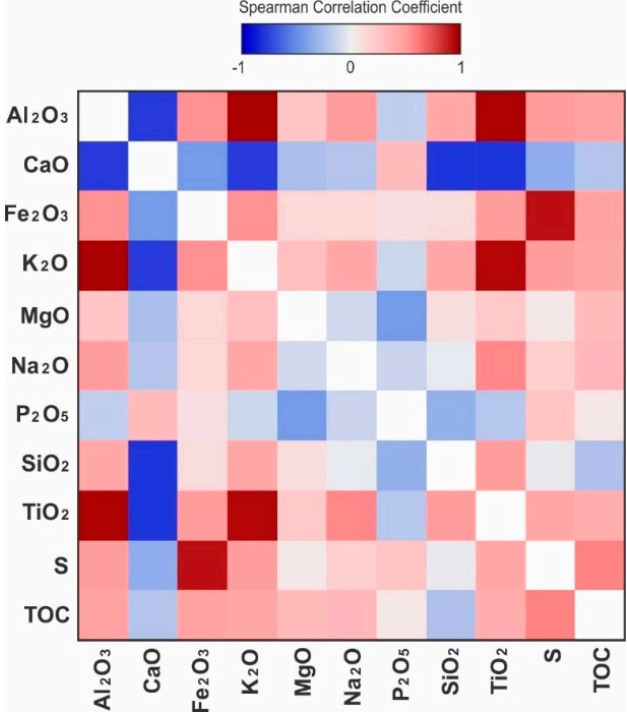

**Figure 13.** Heat map of the Spearman correlation coefficients between all major elements and TOC from Rock-Eval pyrolysis.

Titanium oxide ($TiO_2$) is mostly strongly correlated to feldspar and to a lesser degree, plagioclase, and illite (Figure 14). Among other major elements, $TiO_2$ has a strong positive correlation with $K_2O$, $Al_2O_3$, and subordinately with $Na_2O$. Although according to Smith [45], Ti is known to enter the structure of feldspars, the concentrations from various analyses of feldspars compiled by Smith [45] are well below the bulk concentrations found in these mudrocks and in the average shale [43]. Therefore,

in spite of the correlation, it is not reasonable to attribute $TiO_2$ mainly to K-feldspars in a depositional context. The concentration of $TiO_2$ may be related to the precipitation of titanium oxides in dissolution voids of detrital feldspars [46] during diagenesis. The positive correlation of $TiO_2$ with illite is also likely due to the occurrence of Ti in the structure of illite, or as an associated oxide mineral phase [47]. Illite, plagioclase, and K-feldspar also have a strong to moderate positive correlation with $K_2O$ and $Al_2O_3$ (Figure 14), both of which are related to the presence of these cations in their structure.

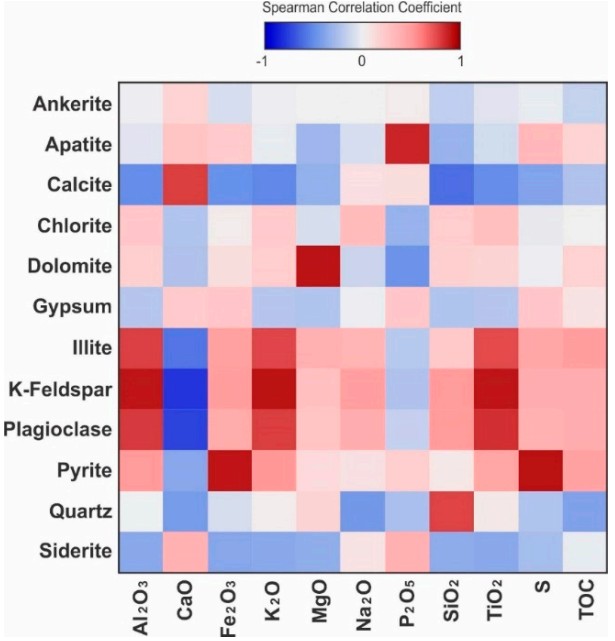

**Figure 14.** Heat map of the Spearman correlation coefficients between mineralogy and major elements plus TOC from Rock-Eval pyrolysis.

Phosphorus oxide ($P_2O_5$) has a poor positive correlation with CaO. The correlation between $P_2O_5$ and CaO is not only due to the carbonate in the apatite structure, but also to the calcite cementation which fills the large interstitial voids between the apatite granules. The relationship between $P_2O_5$ and other elements, including Ca, is not monotonic and is obscured by the bimodality of $P_2O_5$, which occurs as a major constituent in phosphorites and as an accessory in mudstone and siltstone beds. The correlation between $P_2O_5$ and CaO is stronger above a concentration of 8 wt.% $P_2O_5$, as the granular phosphorites allowed pervasive calcite cementation.

There is a weak negative correlation between $P_2O_5$ and magnesium oxide (MgO) in the phosphate-poor samples. The phosphate-rich samples are particularly poor in MgO, and show a stronger negative correlation, which implies that there was no substitution of $Ca^{2+}$ by $Mg^{2+}$ in the apatite structure. The MgO concentration strongly correlates to and is predominantly due to the abundance of dolomite, which followed the calcite cementation in the diagenetic sequence. The low concentration of MgO in the phosphatic samples is likely a result of pervasive calcite cementation, which allowed little space for the precipitation of interstitial dolomite later in the diagenetic sequence. The only significant correlation of MgO with another major element is a weak negative correlation with CaO, which supports this hypothesis. Illite also has a moderate positive correlation with MgO due to the presence of Mg in its crystalline structure [48,49], although Mg may also be from other clay minerals present in trace amounts and undetected by XRD.

Besides the moderate positive correlation of CaO with calcite and subordinately with apatite, due to the incorporation of Ca into their crystal lattice, CaO has a moderately high positive $r_s$ with siderite. Although siderite is absent in most samples and only occurs in trace amounts in a few samples, it is noticeably associated with the reworked granular phosphate facies, where it can reach up to 0.6 wt.%. The facies in which siderite occurs have extensive calcite cementation, which produces the

apparent positive correlation in relatively siderite-rich facies. The strongest negative correlations of CaO and any minerals are with feldspar, plagioclase, and to a lesser extent with illite. This is explained by the stronger affinity that calcite cementation and phosphatic clasts have with bioclasts and hence the relatively lower abundance of the detrital feldspar and plagioclase, while the negative correlation with illite has to do with the lower degree of calcite cementation in finer-grained clay-rich facies.

There is a moderate positive correlation between $P_2O_5$ and S, which may be due to the association of pyrite with the phosphorite facies, but may also be related to the substitution of $PO_4^{3-}$ by $SO_4^{2-}$ in the structure of apatite [50]. The strongest positive correlations of $Fe_2O_3$ are with pyrite and sulfur. The moderate positive correlation of $Fe_2O_3$ with illite and feldspar is attributed to the occurrence of Fe in the structure of illite, feldspars, and possibly other clay minerals, as a substitution cation [51,52].

### 3.3.2. Sulfur and Organic Carbon

Sulfur only has a moderate positive correlation with TOC (Figure 13). When considering the amount of pyrite and $Fe_2O_3$ together with the sulfur and TOC relationship, a population of samples with higher pyrite and $Fe_2O_3$ content deviate from the normal marine trend of Raiswell and Berner [53]. The normal marine trend follows a linear trend between sulfur and TOC, with an extrapolated intercept near the origin (Figure 15). The formation of pyrite in samples that follow the normal marine trend is related to diagenesis. Some samples deviate from the normal marine trend, showing a poorer correlation between sulfur and TOC. If this trend is extrapolated, it yields a positive sulfur intercept, meaning the amount of sulfur and pyrite is higher than what would be expected for their respective value of organic carbon in the normal marine trend. This positive intercept is due to the formation of syngenetic pyrite, in addition to any pyrite that may have formed during later diagenetic phases.

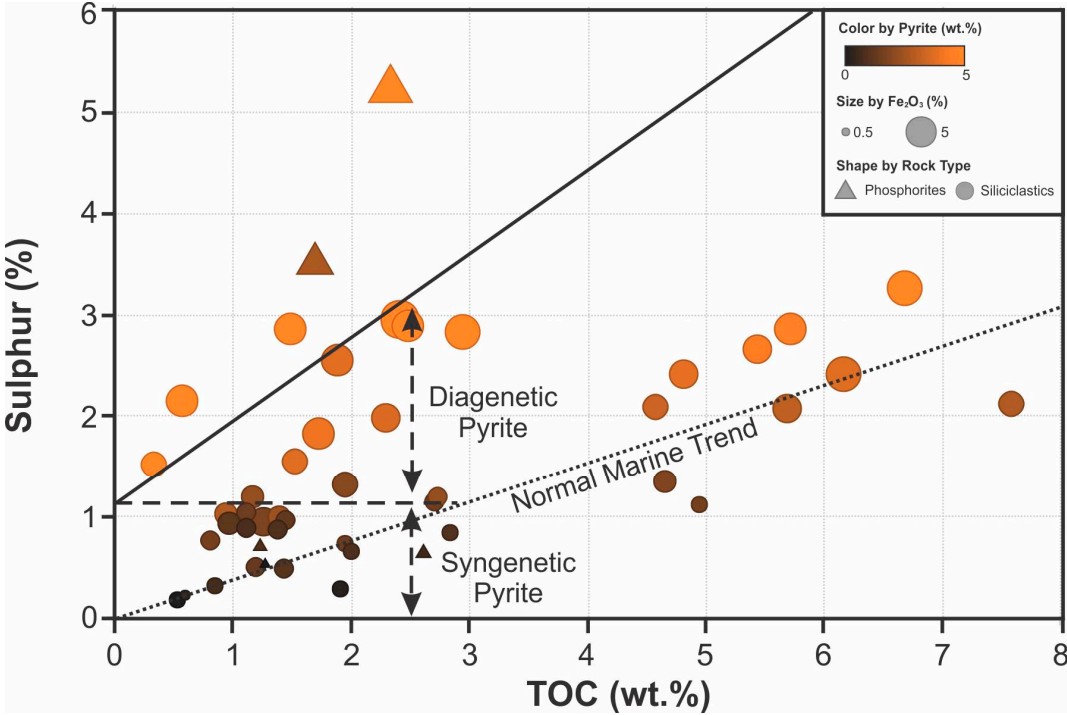

**Figure 15.** Crossplot of sulfur against TOC from Rock-Eval pyrolysis, with marker color by pyrite content, size by $Fe_2O_3$, and shape by rock type, showing two different populations according to pyrite content. Normal marine diagenetic pyrite trendline and a non-zero intercept trendline due to syngenetic pyrite (based on Raiswell and Berner [53]) are shown.

Samples that plot above the normal marine trend (Figure 15), with the additional syngenetic pyrite include the granular phosphatic facies and some samples from the siliciclastic facies. The intraclastic phosphorites, however, show no deviation from the diagenetic pyrite trend. Although initially

considered to represent euxinic bottom-water conditions by Raiswell and Berner [53], the diagenetic pyritization may not necessarily reflect oxygen levels in the water. Instead, the formation of pyrite associated with these phosphorites may be related to the production of sulfide in the sediment column along the ZOP. For example, Łukawska-Matuszewska et al. [54] found that the maximum concentration of iron sulfide in the Baltic Sea shelf occurs 30 to 50 cm below the sediment-water interface. During episodes of phosphogenesis, the long exposure time and the alternation between euxinic conditions in the ZOP and enhanced oxidation [55,56] caused by removal and reworking, and accompanying migration of the ZOP towards the sediment-water interface, are the primary controls on the formation of pyrite and associated sulfides. The trace amounts of siderite associated with the intraclastic phosphate-rich facies are likely sourced from the oxidation of pyrite during the reworking of the underlying granular phosphate beds. The narrow redox stability range of pyrite is bounded by a siderite stability region [57]. The higher energy environment which caused the reworking of the phosphate-rich beds may have caused the partial oxidation of pyrite and supplied Fe for carbonate precipitation.

### 3.3.3. Trace Elements

The Al-normalized enrichment factors (EF) of trace elements relative to the average shale values of Wedepohl [43] were calculated according to the equation in Tribovillard et al. [9]. The relative enrichment of Ce compared to its neighboring lanthanides, La and Nd, was also calculated according to the equation of de Baar et al. [58]. This relative enrichment is known as the Ce anomaly, which is a consequence of the occurrence of the relatively insoluble oxidized $Ce^{4+}$, in addition to the trivalent state in which most of the other REE are found [58]. The value of the Ce anomaly is interpreted to reflect a record of oxygen levels in the water at the time of incorporation into the mineral, varying from less than unity (Ce-depleted) for oxygenated waters, to larger than unity (Ce-enriched) for anoxic conditions. Due to the low amount of Al in both the phosphorite and the siliciclastic facies of the DPZ, the EF of most elements are higher than unity (Figure 16). There are no significant differences in the EF of Cs, Ga, Hf, Li, Nb, Rb, Ta, W, and Zr between the siliciclastic and the phosphorite facies, and these elements correlate strongly with Al or Si (Figure 17).

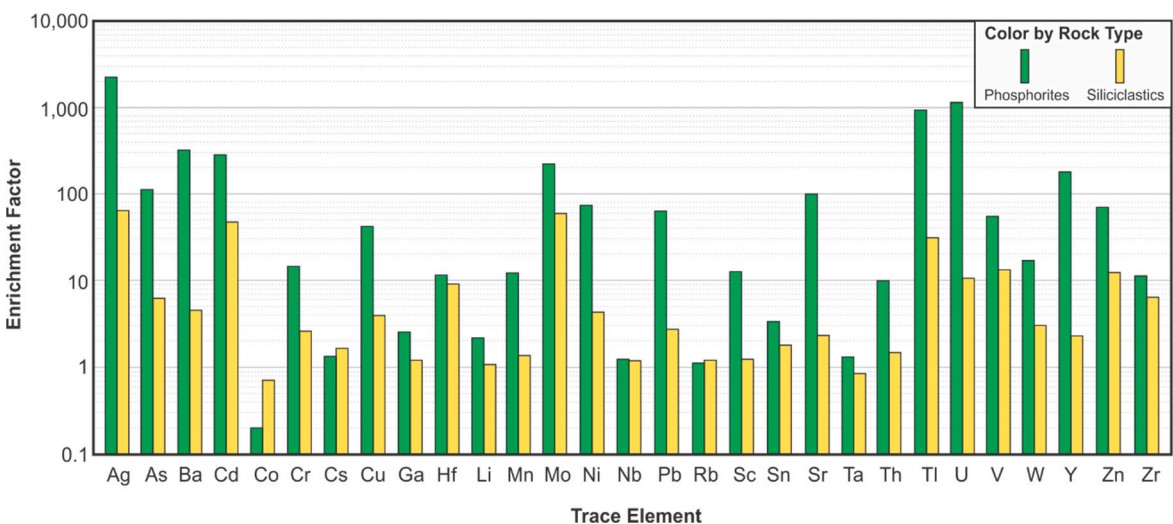

**Figure 16.** Average enrichment factors of trace elements relative to the average shale values of Wedepohl [43], for the phosphorite and the siliciclastic facies of the DPZ.

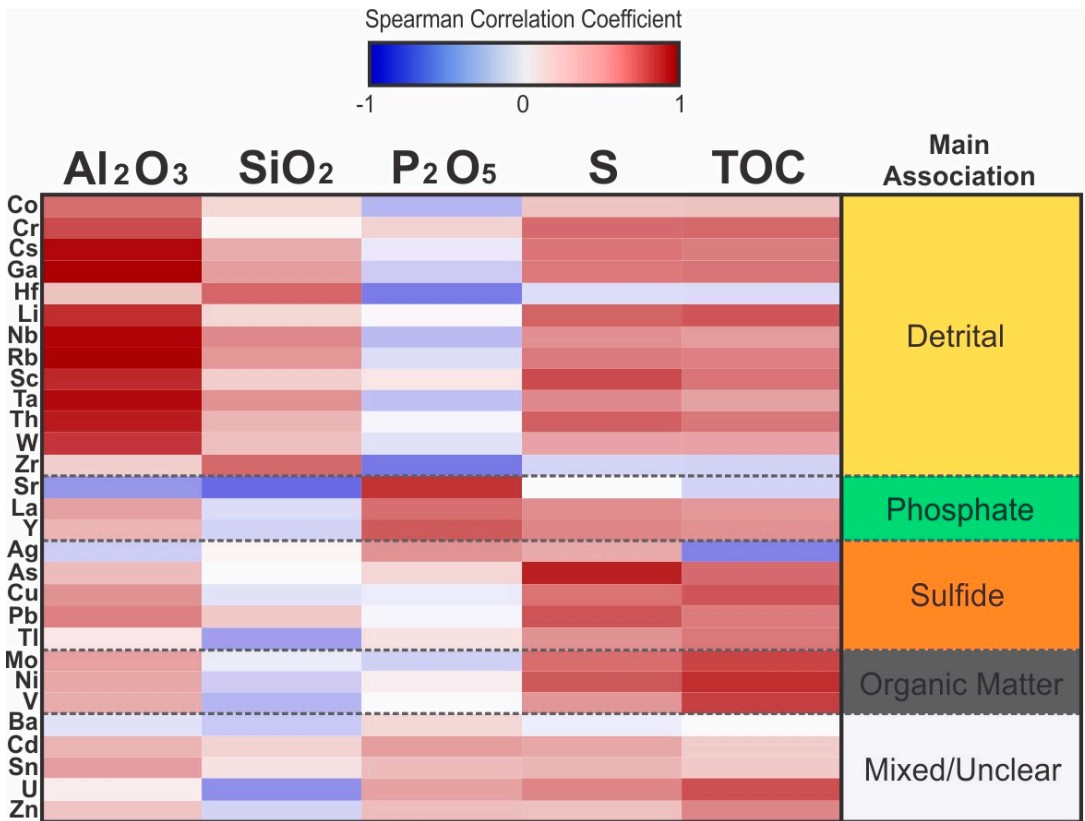

**Figure 17.** Heat map of the Spearman correlation coefficients between select trace elements and major elements plus TOC from Rock-Eval pyrolysis, organized by the association of element with the detrital fraction of the sediment, phosphate, sulfide, organic matter, and mixed or unclear influence.

The elements Cs, Ga, Nb, Rb, and Ta have a strong positive correlation with Al and feldspar, and subordinately with illite. The trace elements Li, Ta, and W also correlate well with Al, illite, and orthoclase, despite precision-related issues due to the presence of these trace elements in concentrations very close to the analytical resolution. The positive correlation of Al with Cr and Sc is also strong, suggesting these trace elements are also mostly related to the detrital fraction of the sediment. The trace elements Hf and Zr correlate with quartz instead of correlating with Al and aluminosilicates (Figure 17), and are also mostly associated with detrital components of the sediment. The relationship between Zr and Hf and quartz is likely due to the association of high specific gravity detrital zircon with the relatively coarser sand and silt-rich facies, since Hf can occupy the Zr-sites within the zircon structure [59]. All samples, and especially the phosphorites, are poor in Co, which has a moderate relationship with Al, suggesting the low concentration of Co is mostly associated with detrital components of the sediment.

The highest trace element EF observed in the DPZ, particularly in association with the phosphorite facies, are of Ag (EF > 2000), U (EF > 1000), Tl (EF > 900) and Ba (EF > 300), Cd and Mo (EF > 200), As and Sr (EF > 100). The EF of Cr, Cu, Mn, Sn, Ni, Pb, Th, V, and Zn are between 10 and 60. The REE of the lanthanide series have EF between 50 and 100, while the EF for Y is 180 and for Sc, 12. The EF of Sr correlates to apatite abundance and reflects the well-documented substitution of $Sr^{2+}$ for $Ca^{2+}$ [50,60]. With the exception of Ce, the lanthanides, as well as Y, have a strong to moderate positive correlation with apatite, due to the substitution of $REE^{3+}$ in $Ca^{2+}$ and $P^{5+}$ sites in the apatite structure [50,60–62]. The Ce anomaly [58] relative to the average shale of both siliciclastics and phosphorites is less than unity, which implies that anoxic conditions were not prevalent. The average Ce anomaly for phosphorites is 0.28, lower than the average of 0.67 for the siliciclastic facies (Figure 18).

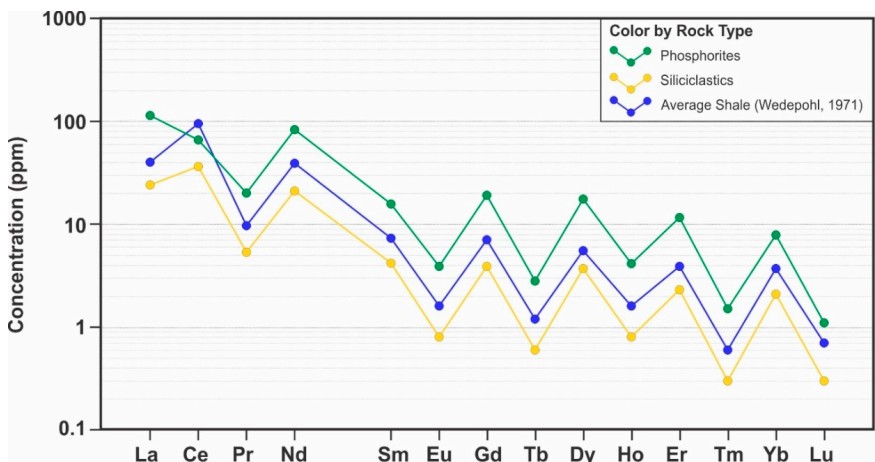

**Figure 18.** Profiles of the average concentration of REE for the phosphorite and siliciclastic facies, and the average shale [43], showing overall REE enrichment and relative Ce depletion in the phosphorites.

In addition to the correlation with REE and Sr, apatite also has strong positive correlations with U and Ba; however, the interpretation is more complex. The substitution of $U^{4+}$ in $Ca^{2+}$ sites [63,64] accounts for the U anomaly in the phosphorite facies, although a distinct relationship with TOC is observed for concentrations of much lower magnitude, due to the chelation of U with organic matter. The solid solution between $Ba^{2+}$ and $Ca^{2+}$ in the structure of fluorapatite [65] may account for its high EF in the granular phosphorite facies, but the average Ba concentration in the intraclastic phosphorites is much lower, without any apparent cause. While Th is largely related to detrital fractions of the sediment (Figure 17), as suggested by its positive correlation with Al, phosphorite facies samples have a separate trend, probably due to the substitution of $Th^{4+}$ in $Ca^{2+}$ sites [63]. The overall concentration of Sn is low and below the detection limit in most phosphorite samples, and there is no clear correlation with any mineral phases or other elements. The presence of Mn is most closely correlated to dolomite, and attributed to its occurrence as a trace element in this mineral, which has been theorized by Rosenberg and Foit [66] and documented by Wen et al. [67].

Several chalcophile elements such as Ag, As, Cu, Pb, and Tl show strong to moderate positive correlation with S (Figure 17) and pyrite. Although these elements have high EF in the phosphorite facies, they have no correlation with apatite, and the concentration of Ag and Tl in the intraclastic phosphorite samples is below the method detectability threshold of 0.5 ppm. According to Jarvis [50], Ag, As, Cu, and Sc are elements typically enriched in phosphorites relative to the average shale; however, their strong association with sulfur and the granular phosphorites, but not the intraclastic phosphorites, implies they are not directly associated with the phosphate. Instead, they may be a product of the long exposure times and sulfidic pore waters, which promoted their incorporation into the sulfide phase. The low concentration of Ag, As, Cu, and Sc in the intraclastic phosphorite facies may be due to winnowing out and remobilization during reworking and redeposition. Sulfur also correlates with Cd and Zn, although there are a few samples that deviate from this trend, with increasing S content while holding approximately the same low concentration of 1 to 2 ppm of Cd and 20 to 30 ppm of Zn.

The concentrations of Mo, Ni, and V are largely correlated to TOC (Figure 17), and are the only trace elements that may be used as proxies of organic productivity and preservation in the DPZ; however, Ni also has a strong positive correlation with S, reflecting its partial incorporation into pyrite [68,69]. Enrichment in Ni, hence, indicates a high organic matter flux, but also the long exposure time necessary for the trapping of released Ni from organometallic complexes and fixation within the sediments, which is usually linked to suboxic and euxinic conditions [9]. The difference in EF of Mo and V between siliciclastic and phosphorite facies is relatively small, and is mainly an Al-normalization artifact due to the low Al content of phosphorites. The phosphorites are not particularly enriched

in Mo and V in absolute terms, and thus these facies are not necessarily associated with enhanced primary productivity, or at least preservation of organic matter.

Processes of scavenging from sea and pore waters and incorporation into sediment are responsible for the enrichment of Mo and V [9]. In the case of Mo, scavenging is promoted by organic detritus or Fe-Mo-S clusters nucleated from metal-rich particles [70,71]. Scavenging of V occurs as the reduced $V^{4+}$ is adsorbed to the sediment surface or incorporated into organometallic ligands [72,73]. Owing to the long exposure time and multiple reworking episodes the phosphorite beds were subjected to, constant exposure to oxidizing waters may have prevented the concentration of organic matter and associated proxy metals in phosphorites. The apatite-bearing siltstones interbedded with phosphorites, have higher concentrations of TOC and the organic matter proxy metals Mo, V, and Ni, than the phosphorite beds.

## 4. Discussion and Conclusions

Apatite occurrence in the DPZ is not evenly distributed, but concentrated in discrete 10 to 20 cm thick-coated grain phosphate grainstone and intraclastic phosphorite beds, or sparsely scattered in interbedded bioturbated calcareous siltstone and mudstone beds. Phosphorite beds are located near the base of the DPZ, and their apatite content varies between 40% and 80% by weight. In the interbeds of siltstone and mudstone facies of the DPZ, apatite is present as grains in amounts of less than 7% by weight. Relative to the average concentrations of major elements in shales, all DPZ sediments are overall enriched in P, as well as Ca and Mg, due to bioclastic components and carbonate cementation. The phosphorite beds are records of condensation due to low sedimentation rates and repeated reworking episodes, which facilitate phosphogenesis by allowing the buildup of phosphate in the pore water.

The DPZ was deposited during a major transgression and represents a period of stratigraphic condensation in the basin; however, the degree of stratigraphic condensation was laterally variable, as previously determined by biostratigraphic analysis [10], and as evidenced by the drastic lateral variation in thickness of the DPZ shown in this study. Paleotopographic lows, possibly caused by localized subsidence controlled by the transition of the basin towards a more tectonically-active setting, allowed the accumulation of thicker sedimentary sections in embayments in the southwestern and central-western portions of the basin. The occurrence of apatite grains throughout the entire lateral and stratigraphic range of the DPZ suggests that phosphogenesis was an active process across the basin. The concentration of apatite in phosphorite beds was controlled by the different sedimentation rates, and phosphogenesis only happened in areas where the sedimentation rate was low enough to allow phosphate buildup in the pore waters through multiple episodes of phosphogenesis and mechanical concentration by reworking.

There is no unequivocal evidence for the origin of the phosphorous based on the findings of this study; however, it is likely associated with the coastal upwelling phenomenon suggested by Davies [18] for the DPZ and often invoked to explain modern and ancient phosphorite deposits on continental shelves, such as the Permian Phosphoria Formation in the northwestern United States [1,74,75]. Water anoxia, however, remains a point of debate as it relates to phosphogenesis; while suboxic to anoxic conditions in the sediment pore waters appear to be a requirement for both microbially induced and inorganic phosphate release and precipitation [5], global ocean anoxic events do not correlate with the distribution of phosphorites [1]. Furthermore, various geochemical indicators in phosphorites suggest at least a partially oxidizing environment or alternating oxic and anoxic conditions [74]. The Ce anomaly values do not support prevailing anoxic conditions for the DPZ, and instead, suggest the phosphorites may have been deposited under more oxygenated waters than the associated siltstones. Fragments of a diverse benthic fauna associated with phosphorites also dismiss anoxia.

The apatite coated grains consist of irregular and erosionally-truncated cortices, which are interpreted to be a result of various phases of phosphatization, exhumation, erosion, and reburial. These processes are repeated through episodes of reworking, winnowing and redeposition in alternating

quiescence and storms or bottom currents, forming unconformity-bounded apatite grains. Some grain truncation surfaces may be partially related to lower-order sequence stratigraphic surfaces within the transgressive phase of the third to fourth-order cycle [16,17] during which the DPZ was deposited; however, the majority of individual grain truncation surfaces are likely related to higher frequency variations caused by seasonal storms and currents. Establishing the precise nature of grain truncations and defining sequence stratigraphic surfaces within the DPZ is further complicated by the condensed nature of the interval. The coated grains were nucleated from aluminosilicates and quartz grains, particles of organic matter, and skeletal fragments. Phosphatic intraclasts with silt-size aluminosilicates and quartz grains contain records of multiple episodes of phosphogenesis and reworking, as evidenced by internal rims and apatite overgrowths. Organic matter is not directly correlated with apatite, but in phosphorites, the organic matter often occurs as solid bitumen concentrated around apatite grains or disseminated within the apatite, as relicts of late-stage oil migration and later alteration to solid bitumen.

Due to the low abundance of detrital components, the Al-normalized EF can be misleading as a representation of abundance in phosphorites and carbonate-rich mudstones. The lithogeochemistry of phosphorites and associated facies, especially of trace elements is influenced by a complex interplay between phosphatization, diagenesis, organic matter primary productivity and preservation, euxinic conditions in the ZOP, and detrital contribution. Rank-order correlation coefficient analysis of major and trace elements allow disentangling of some of these influences. The trace elements Cs, Ga, Hf, Li, Nb, Rb, Ta, W, and Zr are hosted by minerals that are mainly detrital in origin. The elements associated with the apatite are Sr, which substitutes in $Ca^{2+}$ sites, and the lanthanides and Y, which substitute in $Ca^{2+}$ and $P^{5+}$ sites. The phosphorites are also enriched in U, which likely substitutes in $Ca^{2+}$ sites. The U abundance also has a relatively smaller association with the organic matter due to chelation. The relationships of Ba, Th, and Sn with apatite are less clear, due to the competing detrital and diagenetic influences and different trends for intraclastic versus granular phosphorite facies.

Pyrite and the associated chalcophile elements Ag, As, Cu, Pb, and Tl are common near the base of the DPZ in both phosphorites and siltstones. Two different generations of pyrite are recognized by distinct relationships with TOC and crystal habits. Diagenetic pyrite is characterized by a more linear relationship between S and TOC with an intercept near the origin, and larger euhedral crystals surrounding the borders of apatite grains and intraclasts. Pyrite formed syngenetically with phosphatization is characterized by excess S for a given value of TOC, resulting in a non-zero intercept on a bivariate plot of S vs. TOC, and small framboidal crystals disseminated within grains. Pyrite and chalcophile trace element occurrences are not tied to phosphorite beds, but may instead, be related to the long exposure times and sulfidic pore waters during phosphatization.

Due to the low clastic input during the deposition of the condensed section of the DPZ, there may have been insufficient Fe input to remove all the excess sulfur. The excess sulfur that is not removed by pyritization has the potential to be incorporated into kerogen, giving rise to the formation of sulfur-rich Type II-S kerogen [76]. Part of the sulfur in the kerogen may be released as hydrogen sulfide during catagenesis, while a fraction remains in the oil generated [77]. This is arguably related to the high reported $H_2S$ associated with gas production from the Doig Formation and implies a Type II-S kerogen Type for the basal section of the DPZ. Another important implication of the abundance of sulfur in the kerogen is that petroleum generation starts at lower temperatures than for S-poor kerogen [78].

The concentrations of the trace elements Mo, Ni, and V correlate mainly to TOC, and their use as proxies for organic productivity and preservation [9,70–73] is supported. The highest concentrations of Mo, V, Ni, and TOC occur in apatite-bearing siltstones interbedded with phosphorites, rather than in the phosphorite facies. This implies that phosphorites are not associated with enhanced productivity and preservation of organic matter, possibly due to the extensive organic recycling promoted by biological activity during the long exposure times of these sediments.

The phosphorites of the DPZ share features in common with many marine phosphorites. The unconformity-bounded coated grains of the DPZ are also documented in the Oligo-Miocene Timbabichi Formation in Mexico, the Cretaceous Alhisa Phosphorite Formation in Jordan, and modern sediments offshore Peru [38]. Like the Permian Phosphoria Formation in the western United States and the Tertiary Bone Valley Formation in Florida, the Cretaceous and Paleogene Abu Tartur and Nile Valley deposits in Egypt, the Gafsa-Metlaoui Basin in Tunisia, and the Ganntour Basin in Morocco [50], the DPZ phosphorites are depleted in Ce, which are indicative of oxidizing open waters. The DPZ sediments were deposited under similar conditions common to other marine phosphorites, and well documented in the Phosphoria Formation; that is, on a western continental shelf with an open connection to the ocean, during a transgressive stage with little terrigenous input.

**Author Contributions:** Article conceptualization and methodology development by P.L.S. and R.M.B.; data analysis, investigation, validation and original draft writing by P.L.S.; reviewing, editing, supervision and funding acquisition by R.M.B. All authors have read and agreed to the published version of the manuscript.

**Funding:** This research was funded by Geoscience BC, Canadian Natural Resources Limited, Chevron Canada Limited, Devon Energy Corporation, EnCana Corporation, geoLOGIC Systems Ltd., Husky Energy Inc. and AGAT Laboratories.

**Acknowledgments:** The authors would like to acknowledge the generous donation of software by geoLOGIC Systems Ltd., Emerson E&P Paradigm, and TIBCO. The authors would also like to acknowledge the assistance of Gethin Owen, Technical Director of Electron Microscopy at the University of British Columbia Centre for High-Throughput Phenogenomics, and Stuart Sutherland, Martyn Golding, Christiano Ng and Jeanine Grillo for the assistance with the recognition of skeletal grains, and Gustavo Kenji Lacerda Orita for the assistance with petrography.

**Conflicts of Interest:** The authors declare no conflict of interest. The funders had no role in the design of the study; in the collection, analyses, or interpretation of data; in the writing of the manuscript, or in the decision to publish the results.

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
