# Peer review of "Significance and Distribution of Apatite in the Triassic Doig Phosphate Zone, Western Canada Sedimentary Basin"

_minerals, doi:10.3390/min10100904_

Round 1

Reviewer 1 Report

Line 44: Allochtons --> Allochtonous (?)

Line 99: "full-pattern pattern" --> please check this expression.

Line 130: 99.4 SiO2 --> 99.4% SiO2

Line: 250: and are most are composed

Lines 312 and 321: (Figure8Error! Reference source not found.)

FIGURES

Fig. 1 --> difficult to read the small letter in over green and blue colours.

Fig. 2 --> difficult to read name/letter over red colour.

Fig. 3 --> Symbols of structures very difficult to read. Should increase the size of Key Legend.

Page 8, line 196: I don't understand "marked on Figure 1" and also "Figure." on line 198.

Figure 5: red star is too small.

Figure 16: letters/words in the right upper square very difficult to read.

Author Response

Thank you for your time in reviewing this paper. Please find below a summary of the changes implemented and authors' comments on your suggestions and recommendations. The revised version of the manuscript with all of the reviewers' comments incorporated is attached. 

Line 44, 99, 130 and 250: Typos in lines 44 (changed ‘Allochtons’ to ‘Allochthonous’), 99 (deleted repeated word ‘pattern’ in ‘full-pattern pattern’), 130 (added ‘%’ after ‘99.4 SiO2’), 250 (deleted the first ‘are’ in ‘are most are composed’) were addressed.
Lines 312 and 321: "Reference source not found" in lines 312 and 321 was an issue in the early version of the pre-print, which was immediately noticed by the author and corrected, but it seems reviewers were sent the uncorrected version.
Line 196: "Cross-section location marked on Figure 1 and key on Figure 3" means that the location of the cross-sections shown in figures 3 and 4 are marked on figure 1, and the key for figure 4 is in figure 3. Reworded that sentence for more clarity.
Fig. 1 and 2: Outline was added to figures 1 and 2 text for improved legibility.
Fig. 3: Legend symbol size was increased in figure 3.
Fig. 5: Increased the size of the star in figure 5.
Fig. 16: Increased the size of the key in figure 16.

Reviewer 2 Report

General comments:

This is a nice paper, illustrating the mineralogical and geochemical signature of a stratigraphic condensation and its vertical and lateral variability.  

There is only one aspect that in my opinion would need to be addressed a bit more widely. As data and interpretations are nicely tied to a stratigraphic succession deposited during a marine transgression and linked to a laterally variable stratigraphic condensation, I believe that a slightly more thorough discussion about these issues would greatly strengthen the research and make it more appealing to a wider number of geoscientists.

In particular, it seems to me quite important to discuss at least briefly the relationships between compositional/mineralogical data and the variable stratigraphic records of the transgression, as the thin, condensed intervals change basinward into a stratigraphically-expanded succession recognized also by several authors in the same area. Some brief description about stacking patterns and sequence stratigraphy bounding surfaces (sequence boundary, ravinement surface, maximum flooding surface) can also help.

For this general situation, the authors may want to cite Posamentier & Allen (1993), the first paper addressing the difference between retrogradation and condensation occurring landward of older regressive topsets vs. thicker and more diluted deposits deposited basinward, in wedges passively ‘healing’ the previous morphologies (alternatively, it can be cited one of the most recent reviews of the sequence stratigraphy discipline, e.g., Catuneanu, 2019). This peculiar setting appears clearly in the study area, documented by Crombez et al (2017, e.g. their Fig. 2), that is a paper already cited in the MS, but that could be used to further develop this concept. Other similar features were described also in other intervals of the Doig Fm (e.g., Willis and Wittemberg, 2000), that the authors may perhaps want to cite as well.

It would be also interesting to link such differences in the condensation features to the Triassic transition toward a more active tectonic setting, as suggested by Gibling et al. (2015), another paper that has been already cited but that could be used to support this important aspect. The South-westward structural dip shown in Fig. 1 seems to indicate a slight rotation with respect to the West-ward thickening shown on Fig. 2, suggesting the interesting possibility that the transgression was enhanced by differential subsidence. This is again in agreement with Crombez et al. (2017), who suggested for the proto-Canadian cordillera a clock-wise rotation during the deposition of the Doig Fm, and a main Eastern sediment source during the deposition of DPZ, actually matching the isopach map of Fig. 2 and the correlations of Figs. 3 and 4, with shallower facies and more abundant phosphatic granules in the Eastern facies belt.  Additionally, in the North they are also associated with a thinner section.

Other examples of condensed vs basinward-thickening transgressive successions in tectonically active shelf margins controlled by local or regional tectonic deformation are documented in a few papers, e.g., west of the San Andreas fault zone (Hogarth et al., 2007, see also Fig. 10 in Le Dantec et al., 2010) and in the Alpine-Apenninic orogenic belts (Rossi et al., 2018), where the basinward reworking of glaucony-bearing condensed facies took place after an incipient tectonic tilting (see also Fig. 10 d-e in  Rossi and Craig, 2016). Perhaps this behavior could be partly similar to the stratigraphic relationships between thinner and thicker sections, or for wells where phosphatic granules are also associated with phosphatic intraclasts.

Briefly addressing these relationships between stratigraphy, tectonic history and mineralogy/geochemistry  would add a lot of value to the MS, making in my opinion a great paper, of broader international appeal. In addition, it could be good to briefly clarify the frequency and physical/temporal scale of grain ‘truncation’ vs the physical/temporal scale involved in the generation, for example, of the DPZ sequence stratigraphic surfaces (e.g., ravinement surface, maximum flooding surface …) and their respective effects on condensation.

Minor points

  • The figures are of good quality and well readable, it would be just good to mark the trace of the cross sections not only on the structural map of Fig. 1 but also on the isopach map of Fig.2; in relation to the issue discussed in the general comments, this will make more clear the relationships between overall facies, phosphorite bed types and thickness at the scale of the whole study area, favouring the framing of the results into a broader perspective;
  • Concerning the characteristics of organic matter, it would be good to add some information also on Hydrogen Index (if available);
  • The definition (or the interpretation) of some sedimentary structures (i.e., cross-bedding) could be misleading in relation the fine-grained lithologies in which they have been described; it would be also good to provide some information on main sedimentary processes;
  • Sometimes, acronyms are cited without having explained their meaning before.
  • Please find further specific comments in the attached commented pdf.

References

Please find herewith the references discussed in the general comments:

Catuneanu, O., 2019, Model-independent sequence stratigraphy: Earth Science Reviews, 188, 312-388.

Le Dantec, N., Hogarth, L.J., Driscoll, N.W., Babcock, J.M., Barnhardt, W.A., and Schwab, W.C.,  2010, Tectonic controls on nearshore sediment accumulatio and submarine canyon morphology offshore La Jolla, Southern California: Marine Geology, v. 268, 115-128.

Hogarth, L.J., Babcock, J., Driscoll, N.W., Haas, J.K., Inman, D.L., and Masters, P.M., 2007, Long-term tectonic control on Holocene shelf sedimentation offshore La Jolla, California: Geology, v. 35, 275-278.

Posamentier, H.W., and Allen, G.P., 1993, Variability of the sequence stratigraphic model: effects of local basin factors: Sedimentary Geology, v. 86, 1-2, 91-109.

Rossi, M., Minervini, M., and Ghielmi, M., 2018, Drowning unconformities on hinged clastic shelves: Geology, v. 46, p. 439–442.

Rossi, M., and Craig, J., 2016, A new perspective on sequence stratigraphy of syn-orogenic basins: Insights from the Tertiary Piedmont Basin (Italy) and implications for play concepts and reservoir heterogeneity, in Bowman, M., et al., eds., The Value of Outcrop Studies in Reducing Subsurface Uncertainty and Risk in Hydrocarbon Exploration and Production: Geological Society of London Special Publication, 436, p. 93–133.

Willis, A.J., and Wittemberg, J., Exploration significance of healing-phase deposits in the Triassic Doig Formation, Hythe, Alberta. Bulletin of Canadian Petroleum Geology, v. 48, p. 179–192.

Author Response

Thank you for your time in reviewing this paper. Please find below a summary of the changes implemented and authors' comments on your suggestions and recommendations. The revised version of the manuscript with all of the reviewers' comments incorporated is attached. 

Well locations were added to Fig.2, so the cross-section location can be visualized there as well. Cross-section trace was left out to avoid overcrowding.

Added early reference to Fig.3 and 4 when first mentioning the phosphorite beds in 3.Results

The only acronym found that was mentioned without first spelling out was REE, which is now addressed. Rare earth elements is spelled out in the first occurrence and ‘REE’ used thereafter.

The “Error! Reference source not found” error was in the first version of the pre-print, and was reported by the author to the editors very early and corrected. It seems like the reviewers were sent the first version. In the current version this is fixed.

Changed "Doig" to "Doig Formation in line 610.

Added brief description of hydrogen index, kerogen type and thermal maturity when describing range of TOC in subsection 3.3.1 Major Elements and 2.2. Geochemistry

Regarding the cross-bedding in fine-grained lithologies, added: ‘in sandy siltstone beds.’ after ‘…but subordinately cross-bedded’. Cross bedding in sandy siltstones in the Doig was documented by Willis & Wittenberg, 2000 (https://doi.org/10.2113/48.3.179) and Evoy & Moslow, 1995 (https://doi.org/10.35767/gscpgbull.43.4.461)

Added information on sedimentary processes under subsection 3.1 Sedimentology and Mineralogy of the Doig Phosphate Zone, when describing the sedimentary structures.

LINE 562: To address the sequence stratigraphic implications and the main factors controlling the basin evolution and sedimentation, and a more thorough discussion on the relationships between compositional/mineralogical data, marine transgression and laterally variable stratigraphic condensation, the following paragraph was inserted after ‘The phosphorite beds are records of condensation due to low sedimentation rate and repeated reworking episodes, which facilitate phosphogenesis by allowing buildup of phosphate in the pore water’: ‘The DPZ was deposited during a major transgression and represents a period of stratigraphic condensation in the basin; however, the degree of stratigraphic condensation was laterally variable, as previously determined by biostratigraphic analysis [10], and as evidenced by the drastic lateral variation in thickness of the DPZ shown in this study. Paleotopographic lows, possibly caused by localized subsidence controlled by the transition of the basin towards a more tectonically-active setting, allowed the accumulation of thicker sedimentary sections in embayments in the southwestern and central-western portions of the basin. The occurrence of apatite grains throughout the entire lateral and stratigraphic range of the DPZ suggests that phosphogenesis was an active process across the basin. The concentration of apatite in phosphorite beds was controlled by the different sedimentation rates, and only happened in areas where the sedimentation rate was low enough to allow phosphate buildup in the pore waters through multiple episodes of phosphogenesis and mechanical concentration by reworking.’

Regarding the request for clarification on the frequency and physical/temporal scale of grain ‘truncation’ vs the physical/temporal scale involved in the generation of the DPZ sequence stratigraphic surfaces, the following paragraph text was added after ‘These processes are repeated through episodes of reworking, winnowing and redeposition in alternating quiescence and storms or bottom currents, forming unconformity bounded apatite grains’: ‘Some grain truncation surfaces may be partially related to lower-order sequence stratigraphic surfaces within the transgressive phase of the third to fourth-order cycle [16,17] during which the DPZ was deposited; however, the majority of individual grain truncation surfaces are likely related to higher frequency variations caused by seasonal storms and currents. Establishing the precise nature of grain truncations and defining sequence stratigraphic surfaces within the DPZ is further complicated by the condensed nature of the interval.’

LINE 578: On the clarification whether the process forming 'erosionally-truncated cortex' is the same leading to the formation of 'unconfomity-bounded grains', it is stated in the first paragraph of subsection 3.2 that unconformity-bounded coated grains are equivalent to erosionally-truncated grains as described by Pufahl & Grimm, 2003

Reviewer 3 Report

I enjoyed reading this paper and overall feel that, after some improvements, it is a valuable study worthy of publication in the journal 'Minerals'. There are some errors and several aspects should be improved before publication. These are summarized here, with further details provided in the annotated PDF of the manuscript.

The title lacks a meaningful context and geographical location. After 'Zone', insert a comma and 'Western Canada Sedimentary Basin'.

For researchers less familiar with literature on phosphorites, the terms 'unconformity-bounded coated grains' and 'erosionally-truncated phosphatic coated grains' are puzzling, and should be introduced with inverted commas. When reading the abstract, this reviewer misunderstood the first expression and thought that the authors were referring to an unconformity within the sedimentary sequence, as this is the normal usage of the term 'unconformity'.

In the Introduction, the aims of the study are stated to be to understand the role of phosphogenesis in the sedimentation of the western margin of the WCSB. This is a narrow outlook and should be widened. You must consider the global value of your study, not just it's regional significance. Readers of the article outside of western Canada will wish to see how your findings relate to the formation of phosphate deposits elsewhere in the world. The reviewer strongly recommends adding two or three sentences in the Introduction, and extending the final part of the Discussion and Conclusions, to draw out the wider significance of your results. This is vital to increase the significance and value of your paper, and make it worthy of publication in the global journal 'Minerals' rather than in a Western Canadian regional journal.

Under the heading '3. Results', the first paragraph is introductory description and as such is inappropriate under this heading. It should be moved to the Introduction and integrated with the existing text there. Figure 2 should likewise be moved into the Introduction.

Figures 3 and 4 are data-rich, however they do not show instances of the two facies that are described at length in the text. Please indicate instances of these facies on the figures. Ideally, the entire phosphate-bearing sequence should be categorised into facies. However, if this is not easy to undertake, then at least indicate a number of instances of each facies, e.g. using arrows and letters or numbers alongside the stratigraphic logs.

In describing trace element distributions in SEM SDD-EDS elemental maps of their samples, the authors have fallen into the pitfall of spurious results due to spectral peak overlaps. These are for yttrium (overlap with phosphorus) and molybdenum (overlap with sulfur). Text referring to Y and Mo distributions in the images should be deleted. Figures 7 and 11 will need to be modified to remove the SEM images of Y and Mo respectively.

Section 3.3.1 Major elements - this is generally described well and illustrated using appropriate figures; the reviewer has added some further suggestions e.g. on association of Mg with illite (the samples may contain small amounts of Mg-bearing clay minerals).

Section 3.3.2 Sulphur and organic carbon - with Figure 15 - very good representation and interpretation of this data. There is some confusion in the use of terms syngenetic and diagenetic; the authors should check carefully that they have used the correct terms. In Figure 15, the reviewer considers that the non-origin intercept 'trendline' is confusing and unnecessary, and should be removed.

Section 3.3.3 (misnumbered 3.3.2) Trace elements - unfortunately a comparable figure is not provided; this deficiency must be addressed, as the evidence is currently not provided for the associations described by the authors. 'Ce anomaly' is mentioned but this is not linked to the author's dataset (Ce is not included in Figure 16), and no values or graphical representation are provided to illustrate the feature. The organisation of the text needs to be improved, in particular consideration of elements associated with phosphate and with sulfide minerals. It does not make sense to consider in the same sentence a mixture of chalcophile elements such as Ag, As and Cu, and non-chalcophile elements such as Sc, as these are hosted in different minerals. And then is the next paragraph to consider another chalcophile element, Ni, and mention its correlation with pyrite.

Later, in the Discussion section where these associations are reiterated, the lithophile elements Cr and Sc are incorrectly grouped with chalcophile elements enriched in pyrite-bearing samples. The Cr and Sc enrichment of these samples cannot be due to substitution within pyrite; other reasons should be considered by the authors.

In a number of places, elements are referred to as of detrital association, but this is not correct: what the authors mean is that these elements occur within minerals that are detrital components of the sediment. Rewording is recommended.

In describing element-element and element-mineral correlations, in places the authors say these are 'very good': this is ambiguous and the phrase should be replaced by 'strong' and in addition the positive or negative form of the correlation should be stated.

Generally the quality of English is good to very good. The reviewer has highlighted a number of grammatical improvements, particularly where sentence structures are odd and where clarity can be improved. Throughout, the authors use the expression 'relatively to' when they mean 'relative to' - please correct this.

Author Response

Thank you for your time in reviewing this paper. Please find below a summary of the changes implemented and authors' comments on your suggestions and recommendations. The revised version of the manuscript with all of the reviewers' comments incorporated is attached. 

Accepted recommendation of title change and changed it to ‘Significance and Distribution of Apatite in the Triassic Doig Phosphate Zone, Western Canada Sedimentary Basin

Added a sentence about the potential economic significance of the phosphate in the Introduction

The issue with duplicate figures in line 197, after Figure 3, was noticed by the author on the first version of the pre-print and corrected earlier on as per author's request, but it seems the reviewers were sent the uncorrected version. Already resolved.

The “Error! Reference source not found” error was in the first version of the pre-print, and was reported by the author to the editors very early and corrected. It seems like the reviewers were sent the first version. In the current version this is fixed.

‘Relatively’ versus ‘relative to’: Changed as per recommendation.

Correlation qualification: Qualified correlations in terms of strong, moderate and weak, and positive versus negative, as per recommendation.

Correlation quantification: A sentence was added in the beginning of the subsection 3.3.1 Major Elements, quantifying the qualitative assessment brackets of correlation according to their reference rank-order correlation coefficients

Regarding the comment about widening the outlook in the Introduction, the sentence ‘…as well as the general processes and depositional environments of sedimentary phosphate deposits.’ was added after ‘The DPZ provides an opportunity to understand…on the western margin of the WCSB.’

Mapping of the DPZ: Added ‘The Doig Formation, and by extension the DPZ, occur entirely in the subsurface, in the undeformed portion of the WCSB’ to 1.2 Geology; and changed ‘mapping’ to ‘subsurface mapping’ in 1.Introduction, for clarity.

LINE 92: Changed to “correlation of biomarkers”

LINE 98: Added ‘Munson et al.’, previously omitted

LINE 100: Accepted change from ‘Analysis … was’ to Proportions … were’

LINE 103: Accepted change from ‘Geochemical analysis … from 52 samples’ to ‘Geochemical analysis of 52 samples …’

LINE 120: Addressed ambiguity in ‘companion sample’

LINE 127-128: Accepted change in verb tense

LINE 203: Accepted deletion of ‘all’

LINE 225: Accepted correction to ‘leached’

LINE 253: Accepted suggestion of syntax change from ‘is evidenced by some phosphatic coated grains pseudomorphs’ to ‘is evidenced by some pseudomorphs of phosphatic coated grains’

LINE 264: Changed ‘…skeletal fragments show signs of intensive reworking and are mostly unidentifiable.’ to ‘skeletal fragments show signs of intensive reworking and are taxonomic groups are mostly unidentifiable’ for added clarity

LINE 262: Reviewer suggested change from ‘with cortex’ to ‘with a cortex’; however, ‘cortex’ is plural of ‘cortice’ and the sentence refers to multiple cortex

LINE 265: Changed ‘The authors described this grain…’ to ‘The aforementioned authors described this grain…’ to avoid ambiguity and clarify the authors in question are Pufahl & Grimm

LINE 322: Replaced ‘more commonly occurring within coated grains’ by ‘frequently occurring in association with coated grains’

LINE 323: Accepted suggestion to change from ‘pyrite crystals are diagenetically formed’ to ‘pyrite crystals are likely formed during later diagenesis’ in text and associated Fig. 9 caption

LINE 324: Accepted change from ‘syngenetically’ to ‘syngenetic’

LINE 331: Accepted change to ‘in fractures’ in Fig. 9 caption

LINE 372: Changed from ‘…and iron oxide (Fe2O3)…’ to  ‘…and iron (expressed as the oxide Fe2O3)…’

LINE 379: Accepted addition of ‘global’ to ‘… average concentration…’ in Fig. 12 caption

LINE 401: Changed ‘…illite may also be due…’ to ‘…illite is also likely due to…’

LINE 410: Replaced ‘The relationship…is non-monotonic…’ by ‘The relationship…is not monotonic…’ to clarify it refers to a mathematical context and not logic

LINE 450: The authors verified that the statement ‘This positive intercept is due to the formation of syngenetic pyrite, in addition to any pyrite that may have formed during later diagenetic phases’ is correct and there is no confusion related to the use of the diagenetic and syngenetic terms.

LINE 475: Reworded the paragraph introducing the meaning of Ce anomaly to clarify it means the general relative enrichment of Ce when compared to its neighbors.

LINE 477: Removed comma after ‘…trivalent state…’

LINE 496: Changed ‘…and therefore also has an important detrital source.’ to ‘…suggesting the low concentration of Co is mostly associated with detrital components of the sediment.’ to clarify that albeit in small quantities, the correlation with Al is indicative of a detrital source.

LINE 498: Accepted change of ‘The highest EF…’ to ‘The highest trace element EF…’

LINE 499: Accepted insertion of comma after ‘…phosphorite facies…’

LINE 504: Accepted change from ‘REE+3’ to ‘REE3+’

LINE 513: Added ‘average’ before ‘…Ba concentration’ and accepted correction from ‘…are…’ to ‘…is…’

LINE 514: Reworded ‘While Th is largely detrital…’ as ‘While Th is largely derived related to detrital fractions of the sediment …’

LINE 515: Accepted insertion of ‘samples’ after ‘…phosphorite facies…’

LINE 519: Reworded ‘…most closely correlated to, and attributed to its occurrence as a trace element in dolomite…’ as ‘…most closely correlated to dolomite, and attributed to its occurrence as a trace element in this mineral…’

LINE 526: Reworded ‘…are elements that phosphorites are typically enriched in…’ as ‘…

LINE 562-563: Changed ‘The highest concentrations of Mo, V, Ni and TOC are found in apatite-bearing siltstones interbedded with phosphorites’ to ‘The apatite-bearing siltstones interbedded with phosphorites, have higher concentrations of TOC and the organic matter proxy metals Mo, V and Ni, than the phosphorite beds’ for clarity.

LINE 566: Accepted deletion of ‘most of the’ before ‘…modern and ancient phosphorite…’

LINE 572: Reworded ‘…at least partially or alternating oxidizing environments…’ as ‘…at least a partially oxidizing environment or alternating oxic and anoxic conditions.’

LINE 589: Changed ‘unentangling’ to ‘disentangling’

LINE 590: Added ‘are hosted by minerals that are’ before ‘…are mainly detrital in origin.’

LINE 591: Added ‘and the’ before ‘…lanthanides…’

LINE 593: Added ‘likely’ before ‘…which substitutes…’

LINES 593-594: Reworded ‘…has a relatively smaller influence of chelation with organic matter.’ as ‘has a relatively smaller association with organic matter due to chelation.’

LINE 614: Added ‘on a bivariate plot of S vs. TOC’ after’ in a non-zero intercept’

Changed ‘Feldspar’ to K-Feldspar’ in Figure 14

Added explanation of trendlines to Figure 15 caption

Trace Elements subsection numbering resolved

The ambiguity related to the term ‘detrital association’ has been clarified to express that the elements are associated with detrital components of the sediment

Added Figure 17 showing a heat map of the correlation coefficients of trace elements with major elements, and a few references to Figure 17 in the text.

Added Figure 18 showing the rare earth element profiles of the averages for siliciclastic and phosphorite facies compared to the average shale, in which the relative Ce enrichment anomaly can be observed and calculated. Ce and other rare earth elements are not included in Figure 16 due to space constraints and that their concentration is better visualized as a rare earth element profile shown in Figure 18. Added a reference to Figure 18 in the text.

LINE 304: Removed ‘primary’ from ‘…preservation of primary intergranular porosity…’ to avoid confusion

LINE 423: Added citation to ‘presence of Mg in its crystalline structure’ and modified the two instances where Mg is attributed to illite to include the possibility of other clay minerals: ‘although Mg may be also from other clay minerals present in trace amounts and undetected by XRD.’ and ‘attributed to the occurrence of Fe in the structure of illite, feldspars and possibly other clay minerals, as a substitution cation’

LINE 511: Changed ‘…a distinct relationship with TOC exists…due to the chelation of U with organic matter’ to ‘…a distinct relationship with TOC is observed…due to the chelation of U with organic matter’ to avoid ambiguity

LINE 531: Removed ‘…or adsorbed onto the surface of apatite grains.’ after ‘…incorporation into the sulfide phase…’

LINE 573: Moved ‘The Ce anomaly values also do not support prevailing anoxic conditions for the DPZ, and instead suggest the phosphorites may have been deposited under more oxygenated waters than the associated siltstones. Fragments of a diverse benthic fauna associated with phosphorites also dismisses anoxia.’ after ‘Furthermore, various geochemical indicators in phosphorites suggest at least a partially oxidizing environment or alternating oxic and anoxic conditions’ and removed ‘also’ in ‘The Ce anomaly also…’.

LINE 598: Removed the lithophile elements Cr and Sc that were listed in Discussion with the chalcophile elements from the sentence ‘Pyrite and the associated chalcophile elements Ag, As, Cu, Pb, Tl, Cr Sc are common near the base of the DPZ…’ and revisited their mineral association, correlating them instead with the detrital fraction and Al by inserting the sentence ‘The positive correlation of Al with Cr and Sc is also strong, suggesting these trace elements are also mostly related to the detrital fraction of the sediment.’ between ‘…due to the presence of these trace elements in concentrations very close to the analytical resolution.’ and ‘The trace elements Hf and Zr…’

LINE 517: The authors reconsidered the correlations of Sn and replaced the sentence ‘Although Sn has a moderate positive correlation with apatite in the siliciclastic facies, the abundance of Sn is below the detection limit in most phosphorite samples, possibly due to remobilization by pore water and substitution facilitated by the long hiatuses and reworking of the phosphorite beds.’ by ‘The overall concentration of Sn is low and below the detection limit in most phosphorite samples, and there is no clear correlation with any mineral phases or other elements.

The authors disagree with the comment that Ni should be mentioned in the previous paragraph with the other chalcophile elements, instead of in the sentence ‘The concentrations of Mo, Ni and V are largely correlated to TOC…however, Ni also has a strong positive correlation with S…’ Despite being a chalcophile element, in the DPZ Ni is mostly controlled by the organic matter abundance, and only secondarily by sulphides, as it is stated in the second part of the previous sentence; therefore, the authors believe that it should be grouped with the other two metals controlled by organic matter.

Regarding the facies classification in the well logs, the authors consider this is beyond the scope of the work. At the scale the well logs are displayed in order to show stratigraphic correlations and lithological trends, indication of phosphorite facies in the cross-sections is not possible, since the phosphorite beds are 10 to 20 cm intervals.

Spectral interference of phosphorous and sulfur in the EDX yttrium and molybdenum signal: The EDX spectra was reviewed and the Mo signal was deemed unreliable indeed, although the enrichment in Mo is also evidenced in the ICP-MS data. References to the EDX Mo map in text caption and figures were removed. Regarding yttrium, the EDX spectra does identify P and Y in the same phase, in the form of a broad P peak with shoulders corresponding to the Y signal. The ICP-MS data supports that interpretation by indicating a strong affinity of Y and rare earth elements for the apatite phase.

The authors disagree with the comment that the paragraph describing the structural aspects, thickness and log character of the interval, together with Figure 2 (isopach map of the DPZ) should be moved into the introduction. The entire paragraph, as well as the isopach map of Fig. 2, is a result of original work by the authors in mapping the distribution of the phosphate zone, which serves as the groundwork for the work that follows. The first part of the paragraph works as a brief preamble for the general stratigraphic statements of the second part of the paragraph.

The authors disagree with the suggestion for deleting the top diagonal of the symmetrical Spearman correlation heat map on Fig. 13. Removing the top diagonal does not serve any purpose, since it does not save any space, and with the symmetry it is possible to switch from row to column by comparing element with mineral and then sliding across to a different element without having to backtrack, and vice-versa.

Use of inverted commas for ‘unconformity-bounded coated grains’ and ‘erosionally-truncated phosphatic coated grains’ or introduce the term with adequate explanation: The term is not explained in the abstract due to word count constraints, but it is introduced and explained adequately in the main text (Section 3.2 Structure, Composition and Origin of Apatite Grains) with the appropriate reference. The authors do not agree with the use of inverted commas for these terms, since they are neither a novelty nor used in a different way than originally intended.

Erosionally truncated phosphatic coated grains and unconformity-bounded coated grains equivalence: These are equivalent according to Pufahl & Grimm, 2003.

Reviewer 4 Report

The reviewed manuscript is of very good quality with interesting results on the significance and distributions of apatite in the Doig Phosphate Zone, Canada. The authors use a very well-designed methodology which provided all the necessary results that support clearly the conclusions they presented. The English language is adequately used, and no evidence of plagiarism has been detected. 

However, some few points need to be corrected or completed for further improving the manuscript. These are indicated below:

LINE 94: A subsection relative to the materials studied should be added before the various methods. A table with the samples considered in the manuscript should also be given therein.

LINE 487: The inferred Ce anomaly is not shown on the graph of Figure 16. The reader should have visual contact with such an assertion.

LINES 490-552: The authors claim that they observed various correlations between the trace and the major elements as well the mineral content of their samples. However, this piece of information is not accessible in the manuscript. Authors should consider inserting relative figures or tables documenting such correlations.

Other minor points that require further attention and correction are listed below:

LINE 59: Point A’ should be added in the map given in Figure 1

LINE 197: Figure 1 is repeated here. It should be removed

LINE 214: a scale is missing from Figure 5a. Please add it.

LINE 312: remove “Error! Reference source not found.”

LINE 321: remove “Error! Reference source not found.”

LINE 343-344: “Pyrite crystals often occur in association with Mo“ Do the authors mean that they observed high Mo values? This is not obvious from the relative figures. Please clarify better.

LINES 365-366: “overall depleted …….enriched” Do the authors imply that something was removed (depleted) and something was added (enriched)? Please clarify.

LINES 365 – 375: add wt in the concentration values eg. 0.16 wt%

LINE 603: change “phosphorites” to “phosphatization”  -  

Author Response

Thank you for your time in reviewing this paper. Please find below a summary of the changes implemented and authors' comments on your suggestions and recommendations. The revised version of the manuscript with all of the reviewers' comments incorporated is attached. 

LINE 59: Point 'A' was present in Figure 1, but hard to read due to color contrast. Added bold outline to all map text for improved legibility.

LINE 94: Added a paragraph under subsection 2. Materials and Methods, providing a description of core logged and the samples collected, with a summary table.

LINE 197: The issue with duplicate figures in line 197 was noticed by the author on the first version of the pre-print and corrected earlier on as per author's request, but it seems the reviewers were sent the uncorrected version. Already dealt with.

LINE 214: Added scale to Figure 5a.

LINE 312 and 321: "Error! Reference source not found" issue in lines 312 and 321 also seem to only be present in the early version of the pre-print, corrected by the editor as per author's request. The current version of the manuscript does not have this issue.

LINES 365 – 375: wt.% added.

LINE 603: Changed “phosphorites” to “phosphatization”.

LINE 343-344: Sentence about pyrite and Mo association was removed, as the validity of the Mo detection was question by another reviewer and deemed unreliable by the authors.

LINE 487: Added Figure 18 showing the rare earth element profiles of the averages for siliciclastic and phosphorite facies compared to the average shale, in which the relative Ce enrichment anomaly can be observed and calculated

LINES 490-552: Added Figure 17 showing a heat map of the correlation coefficients of trace elements with major elements

LINES 365-366: The depletion and enrichment in question do not refer to removal or addition of elements, but to the presence of elements in concentrations above or below the average concentrations for a reference rock (shale in this case). The authors believe that this is sufficiently clear from the beginning of the sentence ‘Relative to the average concentrations of major element in shales [42]…’.

Round 2

Reviewer 2 Report

I appreciated the description of the relationships with stratigraphic condensation, and the discussion about lower frequency sequence stratigraphic surfaces vs. episodic or seasonal events, although it seems to me that these sentences are a bit generic. Anyway, I understand that this type of discussion could be complicated.

However, addressing more thoroughly such relationships would have strongly increased the value of the paper and especially its more general significance to other similar basins or conditions worldwide (e.g., California, Alps, and possibly others).

Explaining a bit more in detail some of my previous considerations, it would be very good at least to consider that in thinned, more condensed areas in the North-east of the study area, only phosphatic granules occur, whereas to the South, in a stratigraphically more expanded succession the only well showing phosphate-bearing facies is characterized by both phosphatic grains and intraclasts. To me, this is likely linked to the development of a healing phase wedge as documented by Crombez et al. (2017) and/or to tectonic tilting as envisaged by Gibling et al. (2015). In healing phase wedges, in fact, some basinward reworking of shallower elements usually occurs, and even more where this happens in tectonically active settings showing in addition a seaward tilting.

Partly linked to some of the previous considerations, it can be noted that, as also suggested by another reviewer (Reviewer 3), the term ‘unconformity-bounded grain’ may confuse the reader in relation to the normal use of the term ‘unconformity’ (as noted by Reviewer 3, at least using inverted commas would be necessary). I understand that the Pufahl & Grimm’s citation can explain the idea, but it still remains a quite strange terminology that might not comply with what the majority of sedimentary geologists usually mean, and that this concept could create problems due to the very different scale/hierarchy of real unconformities.